



# Global Thermocline Vertical Velocities: a Novel Observation Based Estimate

Diego Cortés-Morales[1,2], Alban Lazar[2], Diana Ruiz Pino[2], and Juliette Mignot[2]

[1]Mediterranean Institute of Advances Studies (IMEDEA), Mallorca, Spain
[2]Laboratoire d'Océanographie et du Climat: Expérimentations et Approches Numériques (LOCEAN-IPSL), Paris, France

**Correspondence:** Diego Cortés-Morales (dcortes@imedea.csic-uib.es)

**Abstract.**

   Vertical velocities at large scales are crucial for understanding ocean dynamics, influencing large-scale circulation and associated biochemical processes, yet their rationale is poorly understood, and their three-dimensional distribution is almost unknown. This paper introduces OLIV3 (Observation-based LInear Vorticity Vertical Velocities), a novel observation-based
estimation product of vertical velocities over the global thermocline. This product relies on the geostrophic linear vorticity balance (LVB) applied to ARMOR3D observation-based meridional velocities with ERA5 Ekman pumping vertical velocity as surface boundary condition. It covers the entire water column, with 1/4° horizontal resolution at annual frequency during the 1993 – 2019 period, available on both depth and isopycnal levels. Since the geostrophic LVB-derived vertical velocities only capture the geostrophic component of the vertical velocity, their performance is tested using an OGCM numerical model
data against the total vertical flow. In the upper ocean, the LVB accurately reproduces the annual variability and captures the climatology of the large-scale total vertical flow (for scales larger than 5○ horizontal resolution) with errors below 50% across the major ocean gyres. OLIV3 capability to estimate vertical velocities in different ocean circulation regimes is assessed against three reference datasets: two reanalysis-based and one observation-based product. A strong spatial and temporal correlation is evidenced between OLIV3 and reanalysis datasets, in contrast to the observation-based product, demonstrating
even higher correlation than within themselves and proving that while the geostrophic components of the reanalyses are highly correlated, the ageostrophic part is not. OLIV3 also reconstructs a baroclinic vertical velocity field, consistent with the basin oceanographic concept of Sverdrup balance theory. Regarding one of the most common applications of vertical velocities, the transfers between the surface and thermocline, results from an OGCM simulation show that the baroclinic geostrophic vertical velocity is a better estimator of the temporal variability of the vertical flow in the ocean interior than Ekman pumping, and
it is essential to consider the variability of the horizontal transport. The OLIV3 dataset developed in this study is available on 50 vertical levels (https://doi.org/10.5281/zenodo.16981061; Cortés-Morales and Lazar, 2025a) and 71 isopycnal levels (https://doi.org/10.5281/zenodo.16962780; Cortés-Morales and Lazar, 2025b).



## 1 Introduction

Ocean vertical motion is fundamental to understanding ocean dynamics. These motions serve as critical mechanisms for the
exchange of properties between the ocean surface and interior, as well as within the ocean interior. The vertical exchanges
encompass essential components such as heat, salinity, $CO_2$, oxygen, nutrients (silicates, nitrates), and contribute to shape the
large-scale thermocline circulation and the Earth's climate regulation (Leach, 1987; Fischer et al., 1989; Klein and Lapeyre,
2009; Mahadevan, 2016; DeVries et al., 2017; Jacox et al., 2018). At finer scales, mesoscale and submesoscale subduction
processes contribute to the ventilation of the thermocline, altering the water mass properties in the ocean's interior, and im-
pacting biogeochemical cycles (Freilich and Mahadevan, 2021). Upwelling motions are also key for sustaining high primary
production by supplying nutrients to the euphotic zone, thereby regulating the biological carbon pump (Freilich and Mahade-
van, 2019; Uchida et al., 2019; Yang et al., 2021). They also constitute one of the key physical processes of the most productive
fisheries regions globally (Pauly and Christensen, 1995). Moreover, the extreme oligotrophy observed in the anticyclonic gyres
is attributed to persistent downwelling vertical velocities (Falkowski et al., 1991).

Although the importance of large-scale vertical velocities ($w$) in ocean dynamics has long been recognised, direct mea-
surements of these motions remain a formidable challenge. This difficulty stems from the extremely weak intensity of the
vertical velocity field relative to large-scale horizontal flows. Near the surface, typical magnitudes are on the order of $10^{-5}$ m
s$^{-1}$, decreasing to $10^{-6}$ m s$^{-1}$ within the thermocline and reaching $10^{-7}$ - $10^{-8}$ m s$^{-1}$ at deeper levels in the ocean interior
(e.g. Stommel and Arons, 1959; Schott and Stommel, 1978). As a result, the basin-scale ocean's vertical flow remains one
of the open questions of physical oceanography. Vertical velocities span nearly four orders of magnitude across spatial and
temporal scales. Observations from Lagrangian neutrally buoyant floats, ADCPs (Acoustic Doppler Current Profilers), and the
Sentinel V ADCP have captured fine-scale vertical velocities with amplitudes ranging from $10^{-2}$ to $10^{-4}$ m s$^{-1}$ (e.g. Bower
et al., 1989; Song et al., 1995; Rossby, 2016; D'Asaro et al., 2018; Comby et al., 2022). Such observations are limited to
small regions (covering often only a few degrees or less) and energetic features that do not reflect the magnitude of large-scale
circulation. Therefore, a combination of observational data and mathematical tools is required to estimate the vertical flow.

Traditional approaches for estimating vertical velocities were derived from tracer fluxes or the application of the continuity
equation to current measurements (e.g. Stommel and Arons, 1959; Robinson and Stommel, 1959; Wyrtki et al., 1961; Munk,
1966; Stommel and Schott, 1977; Schott and Stommel, 1978; Wyrtki, 1981; Roemmich, 1983; Wunsch, 1984; Halpern and
Freitag, 1987; Halpern et al., 1989; Weingartner and Weisberg, 1991). These early methods provided insight into the small
vertical velocities' order of magnitude and upwelling/downwelling patterns of the vertical motions. Vertical velocities have
also been inferred from the divergence of horizontal velocity in numerical models (e.g. Madec et al., 2019), although it remains
impractical for global observation-based applications because of the sparse distribution of direct current measurements. An
exception to such application to observations is Freeland (2013), which used in situ Argo float observations to estimate $w$ at a
single depth in a limited domain of several degrees, assuming zero vertical flow at the surface. In the last decade, alternative
approaches used isopycnal displacements (Giglio et al., 2013; Christensen et al., 2023), the use of mooring data combined
with the momentum and density balances (Sevellec et al., 2015), and biogeochemical tracers (Garcia-Jove et al., 2022). The





theoretical frameworks have also expanded to include methods based on the Bernoulli function to infer $w$ (Tailleux et al., 2023).

A widely used approach for diagnosing vertical velocities is the quasi-geostrophic (QG) omega equation developed initially for the atmosphere by Hoskins et al. (1978). It links the vertical flow to various processes, including the thermal wind imbalance trend, deformation, kinematic deformation, turbulent buoyancy, and turbulent momentum. This diagnostic equation (solvable from a single snapshot) has been used extensively in regional studies (e.g. Tintore et al., 1991; Pollard and Regier, 1992; Rudnick, 1996; Allen et al., 2001; Buongiorno Nardelli et al., 2001; Gomis et al., 2001; Garabato et al., 2001; Rodriguez et al., 2001). However, solving the omega equation requires high-resolution 3D fields and well-defined lateral and vertical boundary conditions. The equation was then updated to account for additional physical processes while maintaining the original Hoskins framework (e.g. Giordani et al., 2006). It was used to assess mesoscale structures in the Atlantic (Ruiz et al., 2014), Southern (Buongiorno Nardelli et al., 2018) and global Oceans (Buongiorno Nardelli, 2020) and front and submesoscale dynamics (Ruiz et al., 2019; Freilich and Mahadevan, 2021; Tzortzis et al., 2021). OMEGA3D (Buongiorno Nardelli et al., 2018; Buongiorno Nardelli, 2020) is the only existing global vertical velocity product based on the omega equation, integrating in situ and satellite-derived fields. While it provides a valuable benchmark, the absence of observation-based ground truth for $w$ underscores the need for alternative approaches and new products to estimate vertical velocities at global circulation scales.

In contrast to the complexity of the omega equation, the linear vorticity balance (LVB; $\beta v = f \partial w / \partial z$) offers a simple diagnostic tool for estimating the geostrophic vertical flow on the $\beta$-plane. When vertically integrated from the surface to a level of no motion, it yields the classic Sverdrup balance, a foundational concept in wind-driven circulation theory (Sverdrup et al., 1947). The recent study of Cortés-Morales and Lazar (2024) has shown in a reference OGCM simulation in the North Atlantic the capability of the gesotrophic LVB-derived vertical velocities to capture accurately the large-scale interannual variability of the vertical velocities, as well as a significant percentage of their time-mean structure, particularly within the thermocline, but also in the intermediate and deep oceans. The LVB's hypothesis breaks down in certain regions. In particular, the approximation is no longer valid near the equator, within the mixed layer, and along western boundary currents where nonlinear processes are no longer negligible in the vorticity balance.

Despite these inherent limitations, this work extends the approach developed for the North Atlantic by Cortés-Morales and Lazar (2024) to reconstruct observation-based geostrophic vertical velocities globally. It thereby delivers the Observation-based LInear Vorticity Vertical Velocities (OLIV3). The work is structured around the following key objectives: (i) Implementation of the LVB framework using geostrophic meridional velocities from ARMOR3D and Ekman pumping from ERA5 as the surface boundary condition (Section 3.1). (ii) Validation and assessment of limitations of the OLIV3 product through an OGCM simulation, treated as a perfect model reference (Section 3.2). (iii) Evaluation of the robustness of OLIV3-derived thermocline vertical velocities in reproducing the known large-scale characteristics of the global vertical circulation with existing observation and model-based estimates (Section 3.3). (iv) Possible physical reasons are proposed to explain why this simplification may or may not be valid in certain regions of the ocean (Section 3.4). (v) Comparison with Ekman pumping estimates to identify regions where LVB simplifications can be valid and necessary for describing the ocean interior (Section 3.5).



## 2 Methodology and Data

### 2.1 Theory for Reconstructing Geostrophic Vertical Velocities

The LVB provides a suitable and well-known foundation for describing the ocean interior flow, as thoroughly described in Cortés-Morales and Lazar (2024). In geostrophic form, it is expressed as:

$$\beta v_g = f \partial_z w_g \tag{1}$$

where $f$ is the Coriolis parameter, $\beta$ is the meridional gradient of $f$, and $v_g$ and $w_g$ are the geostrophic meridional and vertical components of the velocity, respectively. Assuming a geostrophic flow on a $\beta$-plane, this formulation describes a local mass balance between meridional divergent flow and an opposing vertical convergence. Therefore, this formulation allows the estimation of the vertical velocity field in the ocean interior (Charney, 1955; Cortés-Morales and Lazar, 2024):

$$w_g(z) = w_g(z_{ref}) - \int_z^{z_{ref}} \frac{\beta v_g}{f} dz' \tag{2}$$

The geostrophic form of the LVB and the corresponding vertical velocity $w_g$ can be derived from the divergence of the geostrophic flow on a $\beta$-plane (Pedlosky, 1996). Thus, this $w$ can be considered as the geostrophic component of the total vertical flow. One common approach to setting the reference condition assumes a level of no motion, where all velocity components are zero. While some studies support this assumption (e.g. Koelling et al., 2020), a degree of uncertainty remains 105 under real oceanic conditions. Alternatively, satellite observations, with their extensive spatial and temporal coverage, provide an adequate boundary condition. Therefore, it is proposed to use Ekman pumping at the ocean surface ($w_{Ek}$) as boundary condition at the surface.

$$w_{Ek} = \frac{1}{\rho_0} \nabla \left( \frac{\tau^{surf}}{f} \right) \tag{3}$$

Where $\rho$ represents the water density at surface ($1025 \text{ kg m}^{-3}$), and $\tau^{surf}$ the wind stress at the oceanic surface.

### 2.2 Deriving Observation-Based Geostrophic Vertical Velocities (OLIV3)


While Cortés-Morales and Lazar (2024) aimed to illustrate the potential of the LVB framework as a main estimator of the North Atlantic thermocline dynamics, the primary objective of this study is to introduce a new observation-based product of geostrophic velocities derived from the geostrophic LVB. To estimate the global ocean vertical velocities from the divergence of the geostrophic flow, meridional geostrophic velocities from ARMOR3D are used in combination with Ekman pumping 115 vertical velocities derived from ERA wind stress as boundary condition. A summary of the main characteristics of the input





datasets, including physical variables, cover period, temporal frequency, horizontal and vertical resolution is provided in Table 1.

ARMOR3D dataset (Guinehut et al., 2012; Mulet et al., 2012) integrates satellite-derived and in situ observations. Its derivation first involves the construction of synthetic temperature and salinity fields from altimetric surface level anomaly (SLA; 
AVISO+, 2015), and sea surface temperature and salinity (SST and SSS) (Reynolds et al., 2007; Droghei et al., 2018) via linear regression method and covariances calculated from historical in situ observations (EN3 dataset (Ingleby and Huddleston, 2007) and Argo floats (Ollitrault and Rannou, 2013)). The synthetic and observed temperature and salinity profiles are then merged using optimal interpolation (Bretherton et al., 1976). Finally, geostrophic velocities are computed using the thermal wind equation, referenced to the surface geostrophic velocities estimated from the altimetric absolute dynamic topography. The 
mixed layer depth (MLD) is obtained from the minimum of temperature and density threshold equivalent to a 0.2°C decrease. ARMOR3D is available at https://doi.org/10.48670/moi-00052.

Ekman pumping vertical velocities are computed from monthly wind stress data provided by ERA5 (Hersbach et al., 2023), the fifth-generation reanalysis of the European Centre for Medium-Range Weather Forecasts (ECMWF). The ERA5 fields are provided at 0.25° horizontal resolution and can be downloaded from the Copernicus Climate Change Service (C3S) Climate 
Data Store (DOI: 10.24381/cds.f17050d7).

The resulting Observation-based LInear Vorticity Vertical Velocities (OLIV3) product consists of geostrophic vertical velocities derived from ARMOR3D meridional velocities and surface Ekman pumping from ERA5 wind stress, using the integrated geostrophic LVB (Eq. 2). The product spans the 1993 - 2019 period, with a horizontal resolution of 0.25° and available in two vertical grids: 50 vertical levels and 71 isopycnal levels. Both versions are quality-flagged based on the relative er-
ror and interannual correlation coefficient between $w_g$ and $w_{tot}$ in the OGCM perfect model test. The datasets are available on 50 vertical levels (https://doi.org/10.5281/zenodo.16981061; Cortés-Morales and Lazar, 2025a) and 71 isopycnal levels (https://doi.org/10.5281/zenodo.16962780; Cortés-Morales and Lazar, 2025b). A low-resolution version, used in the intercomparison test of this study (Section 3.3), is available upon request.

## 2.3   Existing Estimates of Vertical Velocities

To evaluate OLIV3 performance, we compare the vertical velocities with two reanalyses (GLORYS12v1, ECCOv4r4), and an observation-based product (OMEGA3D). To facilitate comparison, the main attributes of the validation products sources are summarised in Table 1.

Following Cortés-Morales and Lazar, 2024, the reference OGCM simulation for the assessment of validity of the methodology is the Nucleus for European Modelling of the Ocean (NEMO) OGCM OCCITENS run from the OCCIPUT project 
(Penduff et al., 2014; Bessieres et al., 2017, Madec et al., 2019). This simulation is forced by DFS5.2 forcing set, using ERA-Interim and ERA40 reanalyses (Dussin et al., 2016). Neutral density field and the isopycnal surfaces were computed for this study using thermohaline and sea surface height fields applied to Jackett et al. (2006) formulation. The MLD provided by the NEMO OCCITENS simulation is computed using a density criterion of 0.01 kg m$^{-3}$ of density change from the surface following the procedure defined by de Boyer et al. (2004). Outputs from the NEMO OGCM OCCITENS simu-





lation are available upon request (thierry.penduff@cnrs.fr). Geostrophic velocities are derived from the model pressure field (calculated using sea surface height and the hydrostatic equation) using the geostrophic equation via the codes available at https://github.com/meom-group/CDFTOOLS.

The GLobal Ocean ReanalYsis and Simulations (GLORYS; Verezemskaya et al., 2021, Jean-Michel et al., 2021) assimilates via Kalman filter along-track altimeter SLA, satellite SST, sea ice concentration, and in situ temperature and salinity profiles,

using NEMO as the model component (Lellouche et al., 2018). The MLD is defined following the same methodology as in the NEMOS OGCM simulation. This dataset is hereafter referred to as GLORYS12v1 and accessed at https://tds.mercator-ocean.fr/thredds/glorys12v1/glorys12v1_pgn_monthlymeans.html.

The Estimating the Circulation and Climate of the Ocean (ECCO) in its fourth release, version 4 (Forget et el, 2015; Fukumori et al., 2018) employs a 4D-VAR assimilation scheme, integrating satellite altimetry, in situ temperature and salinity

profiles from Argo, satellite sea surface salinity and temperature, and ocean bottom pressure, together with the MIT General Circulation Model (Adcroft et al., 2004). The mixed layer depth is defined following the procedure developed by Kara et al. (2000, 2003), which finds that the optimal estimated of turbulent mixing penetration is obtained with a mixed layer depth definition of $\Delta T = 0.8°$ C. This reanalysis hereafter referred to as ECCOv4r4 is available at https://www.ecco-group.org/products-ECCO-V4r4.htm.

Finally, OMEGA3D is an observation-based global estimate of vertical velocities derived from the QG omega equation (Buongiorno Nardelli et al., 2018; Buongiorno Nardelli, 2020). It is based on ARMOR3D thermohaline field and geostrophic velocities, and ERA-Interim (Dee et al., 2011) surface air-sea fluxes. OMEGA3D is available at https://doi.org/10.25423/cmcc/multiobs_glo_phy_w_rep_015_007.

**Table 1.** Summary of datasets used for OLIV3 computation and validation.

| Dataset | Type | Resolution (hor./vert.) | Period | Frequency | Variables Used |
|---------|------|-------------------------|--------|-----------|----------------|
| ARMOR3D | Obs-based | 0.25° / 50 levels | 1993–2019 | weekly | $v_g, T, S$, MLD |
| ERA5 | Reanalysis | 0.25° / surface | 1979–present | monthly | Wind stress |
| NEMO-OCCITENS | Model (OGCM) | 0.25° / 75 levels | 1960–2015 | monthly | $w_t, w_g, T, S$, MLD |
| GLORYS12v1 | Reanalysis | 1/12° / 50 levels | 1993–2019 | monthly | $w_t, T, S$, MLD |
| ECCOv4r4 | Reanalysis | 1° / 50 levels | 1992–2017 | monthly | $w_t, T, S$, MLD |
| OMEGA3D | Obs-based | 0.25° / 75 levels | 1993–2019 | weekly | $w$ (QG) |
| OLIV3 | Obs-based | 0.25° / 50 levels | 1993–2019 | yearly | $w_g$ |

## 2.4 Validation Methodology

Validating OLIV3 with observational data is problematic due to the lack of a ground truth for large-scale vertical velocities. Here, the performance of OLIV3 relies on the consistency across existing $w$ estimates: GLORYS12v1, ECCOv4r4 and OMEGA3D. The intercomparison is conducted on a common spatiotemporal resolution: annual means at 5° horizontal resolution and isopycnal levels. This choice reduces vertical grid and thermohaline structure differences. It also allows to focus





on large-scale dynamics, which are better resolved by the LVB framework (Cortés-Morales and Lazar, 2024). The isopycnal levels are defined by the neutral density of each dataset (Jackett et al., 2006). For the OLIV3 and OMEGA3D datasets, the thermohaline field used to interpolate $w$ onto isopycnal levels is ARMOR3D, since the velocity field was constructed using it (Buongiorno Nardelli, 2020). Diagnostics are computed over the overlapping 23 year period (1993 - 2015) and include time-mean, time-mean vertical gradient, variance and correlation coefficient (R). Regions within the equator band (5°S/N) are excluded as the geostrophic equation cannot be solved at these latitudes.

## 3 Results and discussion

### 3.1 Observation-based Linear Vorticity Vertical Velocities (OLIV3)

Twenty five year mean geostrophic vertical velocities ($w_g$) stemming from the OLIV3 product at $\sigma 26$ isopycnal surface are presented in Fig. 1. This isopycnal level was chosen to assess the vertical velocity estimates across most of the extension of the global subtropical gyres, while maintaining a focus on thermocline dynamics, where the LVB framework performs best (Cortés-Morales and Lazar, 2024). The geostrophic vertical velocity field (OLIV3) within the tachocline, understood as the upper ocean layer defined by a strong vertical shear in the velocity field (Cortés-Morales and Lazar, 2024), reproduces the well-known wind-driven circulation features, generally with upwelling at tropical latitudes and downwelling at the subtropics. This emphasises the role of atmospheric forcing as the primary driver of vertical flow within the upper ocean (e.g. Huang and Russell, 1994; Liang et al., 2017). The Pacific and Atlantic eastern tropical upwelling systems that continue along the eastern coast up to subtropical latitudes are associated with maximum values near the coast around 2 x $10^{-6}$ m s$^{-1}$ (Aristegui et al., 2009). The anticyclonic circulation of subtropical systems is characterised by negative $w$ (downwelling) with maximum values found along western boundaries. This pattern differs from the agreement among reanalyses about upwelling in most of the western boundary current systems, in particular the Gulf Stream and the Kuroshio Current as demonstrated by Liao et al. (2022). In the Gulf Stream case, this poor agreement with reanalyses in the literature aligns with the lack of confidence in the LVB as an estimator of the vertical flow, as demonstrated by Cortés-Morales and Lazar (2024). Outside the Northern Hemisphere subtropical band, some upwelling occurs over the extension of the Gulf Stream and Kuroshio systems, which can be explained by Ekman pumping (Qiu and Huang, 1995).

To analyse the temporal and vertical variability of the vertical velocity estimates, Fig. 2 shows regionally averaged $w_g$ as a function of isopycnal level and year for three distinct regions in the North Atlantic Ocean: the eastern tropical, subtropical and subpolar gyres (purple regions in Fig. 1). The bottom of the layer was selected as the level where the estimates' climatology changes sign. Figure 2 evidences that the direction of geostrophic vertical velocity estimates maintains their sign throughout most of the layer's thickness. Before reaching this depth, the annual velocities reduce in amplitude, supporting the baroclinic nature of vertical flow in the tachocline found in Cortés-Morales and Lazar (2024). This behaviour is consistent with the requirement of a level of no motion at depth to satisfy the Sverdrup balance (Thomas et al., 2014).

The velocity sign remains unchanged over time in the three regions (Fig. 2), but temporal variability is evident across them. Weaker upwelling events in North Atlantic Tropical Gyre (e.g. 1995, 2001, 2005 and 2010 in top panel in Fig. 2) seem

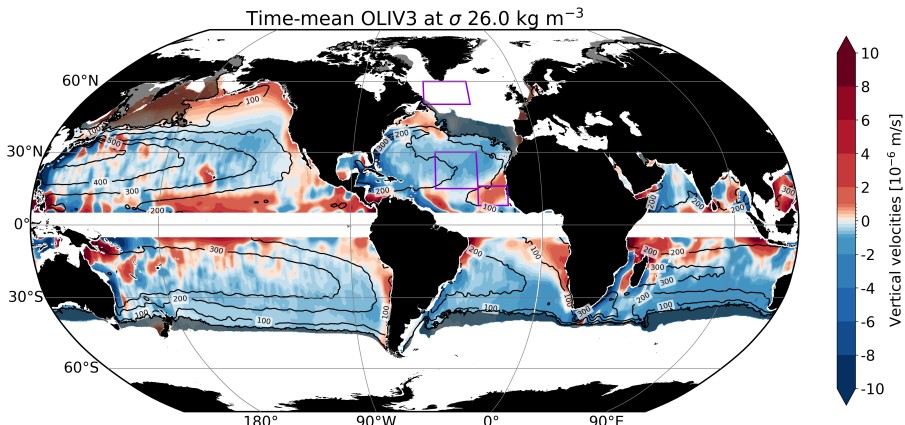

**Figure 1.** OLIV3 time-mean vertical velocity at $\sigma26$ (1993-2018). The field has been smoothed with a 5° running mean. Translucent black shading represents regions within the maximum mixed layer over the study period. The black contour lines represent the depth of the isopycnal surface in meters. Purple rectangles indicate the average regions for Fig. 2.

associated with the negative North Atlantic Oscillation (NAO) phases shown by Pinto et al. (2012) and Roch et al. (2024), while positive NAO phases correspond to stronger upwelling. This suggests that the variability of the eastern tropical gyre is modulated by the phase of the NAO. Interestingly, the magnitude of the oceanic response does not reflect the NAO intensity
presented in the references above. Focusing on the subtropical gyre (central panel in Fig. 2), maximum downwelling events do not correlate with the NAO index as clearly as the tropical upwelling does. Studies such as Pinto et al. (2012) suggest that the atmospheric changes modify both the intensity of the downwelling and the location of the subduction maximum. Therefore, the out-of-phase relationship between subtropical vertical flow variability and NAO index may suggest that the region selected for this study does not fully reproduce the gyre interannual variability. However, other components can influence the variability of
the region as the NAO is not the sole contributor to atmospheric forcing variability (Zhao and Johns, 2014). In the subpolar gyre (bottom panel in Fig. 2), oscillations in upwelling amplitude exhibit a periodicity of approximately 5 years, with particularly strong positive velocity events during 2002-2003, 2007 and 2009. These features align with observed changes in the subpolar gyre from satellite altimetry (Foukal et el, 2017) and volume transport estimates of the East Greenland Current (Daniault et al., 2011). These findings suggest that OLIV3, while capturing the baroclinic nature of the vertical velocity field in the tachocline,
is also capable of transmitting some of the surface interannual signal into the ocean interior.

## 3.2   Linear Vorticity Balance Framework in a Perfect Model Test

The vertical flow computed by applying the Eq. 2 to the ARMOR3D dataset represents the geostrophic component only. Therefore, before comparing OLIV3 with estimates of the total vertical flow, the limitations of this formulation are evaluated using an OGCM simulation, considered as a "perfect model test". This global-scale analysis extends the regional assessment
conducted by Cortés-Morales and Lazar (2024) for the North Atlantic Ocean. The absolute relative error between the time-



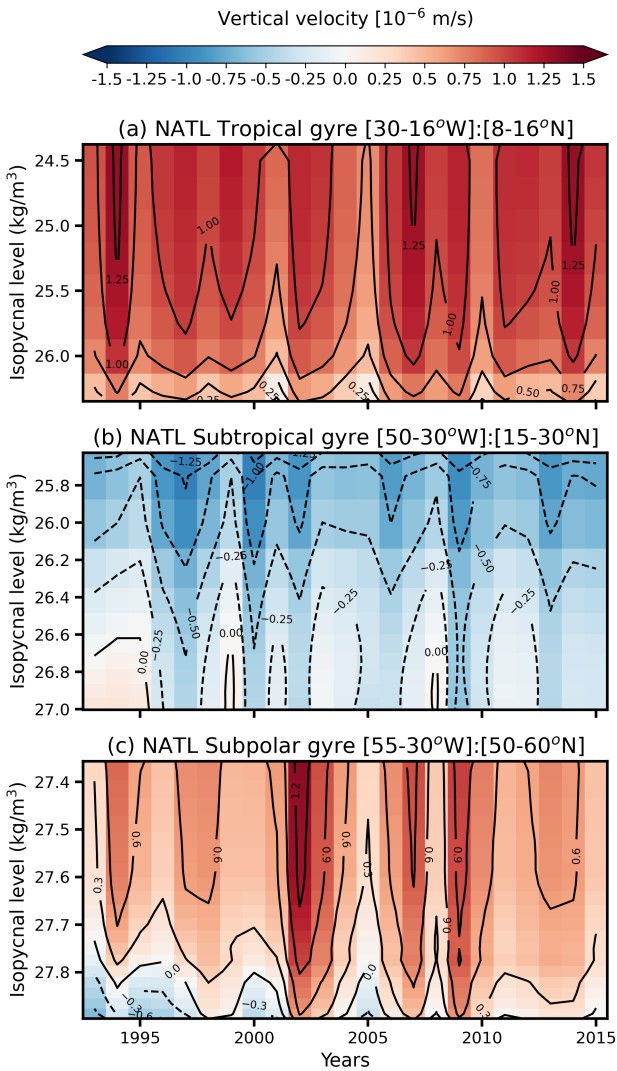

**Figure 2.** Regionally averaged vertical velocity estimates from OLIV3 as a function of isopycnal surface and year for three subregions in the North Atlantic Ocean (NATL): (top) Tropical Gyre (30-16°W, 8-16°N), (middle) Subtropical Gyre (50-30° W, 15-30°N), and (bottom) Subpolar Gyre (55-30°W, 50-60°N). For each region, the first isopycnal level corresponds to the shallowest level that does not intersect with the sea surface. Contours of vertical velocities have been included for readability (m s$^{-1}$).

mean geostrophic vertical velocity ($w_g$) and the total vertical velocity ($w_{tot}$), their interannual correlation coefficient at $\sigma 26$, as well as the relative error in the vertical gradient between the bottom of the mixed layer (MLD) and $\sigma 27$ computed using OGCM over the period 1993 - 2015 are presented in Fig. 3. The vertical gradient of vertical velocities between the base of the maximum mixed layer and $\sigma 27$ has been normalised by the density interval between these isopycnal levels, as follows:



$$\partial w = \frac{|w_{\sigma 27}| - |w_{\sigma MLD}|}{\sigma_{27} - \sigma_{MLD}} \tag{4}$$


Negative values indicate a decrease in magnitude with depth, while positive values indicate an increase. The velocity fields were smoothed with a 5° x 5° running mean to retain large-scale structures that LVB can describe (Cortés-Morales and Lazar, 2024). As shown in Cortés-Morales and Lazar (2024) for the North Atlantic, $w_g$ succeeds in estimating $w_{tot}$ over most parts of the basins. Fig. 3a shows that values are accurate, with a relative error below 50%, across most of the global tropical and

subtropical gyres (yellow, orange and red regions), as well as the eastern part of the subpolar Pacific gyre and the arctic Beaufort gyre. This result shows that the geostrophic vertical velocity field generally reproduces the spatial structure and amplitude of the thermocline vertical flow within the major gyres in the model simulation, suggesting that a similar behaviour may occur in the real ocean.

Relative errors exceed 50% in several regions. These include the intergyres bands, where vertical velocities change sign, as

well as larger regions of intense current systems, such as the Gulf Stream and its northeastward extension, the Brazil-Malvinas Current, the Kuroshio Current, the Eastern Australia Current and the Algulhas Current (see locations in Barceló-Llull et al., 2025). High relative errors are also found in regions dominated by strong zonal flows, like the deep tropics, such as the Equatorial Currents and Countercurrents, but also open ocean poleward extensions of western boundary currents like the North Atlantic Drift (NAD), and the Antarctic Circumpolar Current (ACC), more visible on deeper isopycnals (not shown). They

are characterised by an intense zonal component, which geostrophic part, unlike the meridional component, does not generate divergence and geostrophic vertical velocity (Cortés-Morales and Lazar, 2024). Most eastern boundaries, particularly eastern boundary upwelling systems (e.g. California or Benguela), as well as the northern Indian basins, suffer from high errors in the estimation of their time-mean vertical velocities. Many of these discrepancies typically arise in regions where the LVB (Eq. 1) no longer holds (hatching in Fig. 3a). In such areas, nonlinear processes, friction and lateral diffusion become essential to close

the time-mean vorticity budget (e.g. Sonnewald et al., 2019; Waldman and Giordani, 2023; Khatri et al., 2023).

It is interesting to note that in certain regions, the large differences are not consistent with the local validity of the LVB at the current isopycnal level. For example, in the North Atlantic intergyre region, a large relative difference between $w_g$ and $w_{tot}$ appears as a narrow band around 15°N, despite the LVB terms showing relative agreement within 10% (i.e., outside hatching). In contrast, within the tropical gyre (centred around 10°N), the relative error between $w_g$ and $w_{tot}$ is found to fall

below 50%, yet the LVB is not valid on $\sigma 26$ there (hatching areas). Indeed, as discussed in Cortés-Morales and Lazar (2024), the geostrophic vertical velocity at a given depth is computed via vertical integration of the meridional transport above that level. Therefore, local deviations from the LVB at a given depth, implying that the export of water is not fully conserved, do not necessarily lead to large errors in the integrated geostrophic velocity. This is more pronounced at deeper levels where the amplitude of the meridional transport is reduced due to the baroclinic structure of the tachocline flow.

Considering now the reconstruction of the interannual variability, the LVB method appears to be strikingly accurate (Fig. 3b). Correlation between $w_g$ and $w_{tot}$ exceeds 0.9, and a variance explained at more than 80%, across most of the tropical, subtropical global ocean and northern subpolar gyres (only Pacific subpolar gyre visible), and the Beaufort polar gyre. These





good results extend those previously reported for the North Atlantic in Cortés-Morales and Lazar (2024). Weaker, or even negative, correlation values are found within the major western boundary current systems of the Pacific, Atlantic and Indian Oceans, where nonlinear dynamics are stronger. Similarly large differences also characterise Atlantic and Pacific deep tropics, the open ocean poleward extensions of western boundary currents, and the entire ACC, visible southeast of Africa. These regions are all dominated by flows with strong zonal components relative to the meridional flow, likely generating mostly ageostrophic vertical velocities that cannot be captured by the LVB framework. Remarkably, very high correlations persist even in regions where the time-mean geostrophic component fails to replicate the total vertical velocity pathways (Fig. 3a), as well as in regions where the LVB does not hold, such as the mixed layer (translucent black surfaces in Fig. 3a).

Figure 3c illustrates the ability of time-mean $w_g$ to represent the vertical structure of the velocity field shown in Fig. 3a, by displaying the relative error of a proxy of the vertical gradient (Eq. 4). Note that the vertical gradient of the total vertical velocity is almost everywhere positive (non-dotted areas), corresponding to the downward decrease of current intensity towards the bottom of the thermocline, suggesting that the velocity field is baroclinic and generating a tachocline, underlined in the North Atlantic by Cortés-Morales and Lazar (2024). This vertical gradient appears to be well represented by $w_g$, where the five main subtropical gyres are characterised by relative error smaller than 20%. A downward increase of $w_{tot}$ intensity (dotted regions) is found in western boundary current systems such as the Gulf Stream, Kuroshio and Brazil-Malvinas Currents, the North Pacific and North Atlantic deep tropics, and the intergyre regions between the major tropical and subtropical gyres. Nonetheless, in the Pacific and Atlantic tropical gyres, the errors in the vertical gradient of the total and geostrophic vertical flow are larger than in the subtropical gyres. The spatial distribution of these errors is similar to the pattern of relative errors above 10% between $w_g$ and $w_{tot}$ (Fig. 3a). Therefore, these results suggest the limitations of the LVB framework in reproducing the time-mean value at a given depth and the vertical structure of the total vertical flow at tropical regions and western boundary current systems.

These findings evidence the relevance of the geostrophic LVB framework for capturing and explaining the dynamics of the large-scale vertical motion in an OGCM simulation. Most notably, the high and widespread interannual synchrony between $w_{tot}$ and $w_g$ suggests that the geostrophic component strongly dominates the interannual variability, and therefore likely in the real ocean as well. While nonlinear processes influence the mean amplitude of the vertical flow, they play a smaller role in modulating this variability. Thus, extending previous results from the North Atlantic Ocean to global scales, it shows that geostrophic vertical velocity provides a reliable estimate of the total vertical flow for studying the climatological flow structure within the interior of the major gyres and the interannual variability of vertical motion throughout much of the tropical and subtropical oceans, parts of northern subpolar gyres and polar gyres. This supports the relevance of applying the same reconstruction methodology to observation-based data. In the rest of the paper, we assess our observation-based estimate of the global thermocline vertical velocities, named OLIV3.

### 3.3 Assessment of OLIV3 Relative to Existing Vertical Velocity Estimates Over the Global Thermocline

Due to the lack of direct observations for large-scale vertical velocities, it is necessary to evaluate the performance of OLIV3 relative to commonly used products. In particular, acknowledging the multidimensional nature of the vertical velocity field,


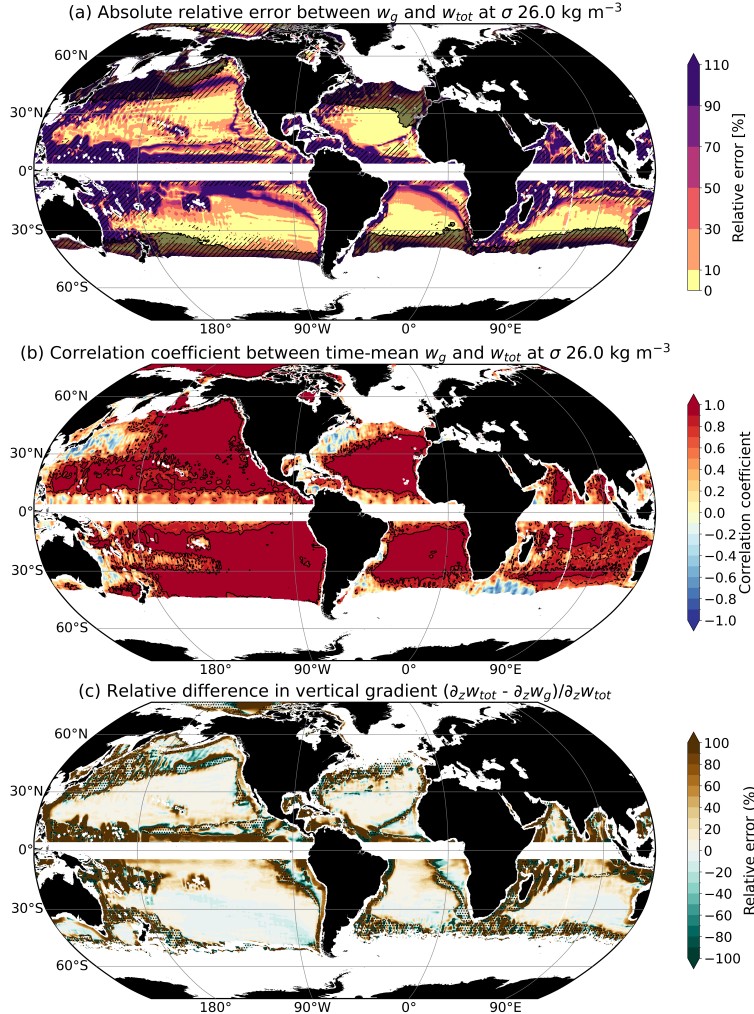

**Figure 3.** Assessment of OGCM geostrophic vertical velocity estimate against the OGCM total vertical velocity. (a) Absolute relative error between means of geostrophic vertical velocity and total vertical velocity at $\sigma 26$ isopycnal surface. Translucent black shading represents regions within the maximum mixed layer over the study period and hatching delimits the areas where the LVB does not hold (relative error > 10%). (b) Correlation coefficient between the annual geostrophic vertical velocity and total vertical velocity at $\sigma 26$ isopycnal surface. Black contours indicate correlation coefficients of 0.7 (dashed) and 0.9 (solid). (c) Relative difference in vertical gradient of vertical velocity between the mixed layer base and $\sigma 27$, computed as $(w_{\sigma 27} - w_{\sigma MLD})/(\sigma_{27} - \sigma_{MLD})$. Dotted (no dotted) areas indicate regions where the total vertical velocity display a positive (negative) vertical gradient, meaning increasing magnitude with depth.

a comprehensive evaluation must address the mean three-dimensional structure, as well as the temporal signals relevant for future studies.





### 3.3.1 Large-scale climatological vertical flow features

The ability of OLIV3 to represent the large-scale climatological upwelling/downwelling structures is evaluated by comparing the 23-year mean at $\sigma26$ against three reference datasets: GLORYS12v1 and ECCOv4r4 reanalyses, and OMEGA3D observation-based product. The large-scale key wind-driven features captured by OLIV3 in Fig. 4 are consistent at first order with those in the reference datasets, with vertical velocity amplitude falling within a common range of values from 0.1 to 10 x $10^{-6}$ m s$^{-1}$.

In the tropics, OLIV3, ECCOv4r4 and GLORYS12v1 (Fig. 4a, c and d respectively) exhibit spatially variable upwelling across all the oceanic basins, including along the Pacific and Atlantic eastern boundaries (Elmoussaoui et al., 2005, Faye et al., 2015). OMEGA3D (Fig. 4b) captures a broader and smoother field with maximum upwelling at the centre of the basins. These OMEGA3D upwelling patterns were already reported in Buongiorno Nardelli (2020). Notably, the deep tropical band (5-10°N/S) show low agreement between reanalyses and OLIV3, matching the regions where the LVB errors in the OGCM

exceed 10% (black hatching in Fig. 4a). Some regions exhibit amplitude discrepancies among the three datasets (e.g. North Pacific tropical band), and some others even display opposite sign (e.g. Indian Ocean).

In subtropical latitudes, OLIV3 and OMEGA3D tend to overestimate downwelling relative to the reanalyses. Despite this overestimation, OLIV3 estimates align more closely with reanalyses than OMEGA3D. Some regional differences emerge, for example, in the North Pacific, where OLIV3 and GLORYS12v1 show a maximum downwelling centred near 30°N, 140°W,

while OMEGA3D and ECCOv4r4 display strong downwelling across the entire 30°N band. In the North Atlantic, the downwelling maximum in OLIV3 and GLORYS12v1 is found in the southeastern part of the gyre. Moreover, OMEGA3D captures this maximum closer to the western boundary current extension and ECCOv4r4 centres it at 30°N. In the South Atlantic and South Indian Oceans, all datasets reproduce maximum downwelling near the intersection of $\sigma26$ with the bottom of the mixed layer. In the South Pacific, only OMEGA3D presents a maximum near the intersection between the isopycnal level and

the bottom of the mixed layer, GLORYS12v1 produces weaker amplitudes, and OLIV3 and ECCOv4r4 locate a maximum downwelling at around 30°S, 90°W.

Western boundary currents also reveal discrepancies. In the Gulf Stream and Kuroshio current extensions, OLIV3 reproduces upwelling patterns consistent with Ekman pumping (Qiu and Huang, 1995), OMEGA3D and reanalyses, while capturing strong downwelling at 30°N along the continental section of both currents. At this latitude, in the Gulf Stream case, OMEGA3D re-

veals some downwelling, while ECCOv4r4 displays positive vertical velocities, in agreement with other model results (GODAS and SODA, among others) shown by Liao et al. (2022). GLORYS12v1 reproduces upwelling and downwelling on both sides of the current. In the Kuroshio Current, ECCOv4r4, GLORYS12v1 and OMEGA3D feature mainly upward flow. The Brazil Current is associated with upwelling flow in all datasets, although OLIV3 tends to overestimate its amplitude.

One may argue that a likely source of discrepancies across the isopycnal level between OLIV3 and the reanalyses is that

OLIV3 reconstructs only the geostrophic component of the vertical velocity, whereas the reanalyses estimate the full vertical velocity field. However, the comparison between OGCM's $w_g$ and $w_{tot}$ in Fig. 3a illustrates better agreement in spatial patterns and intensity than when comparing OLIV3 with the reanalyses in Fig. 4a, c and d. For instance, in the subtropical gyres,

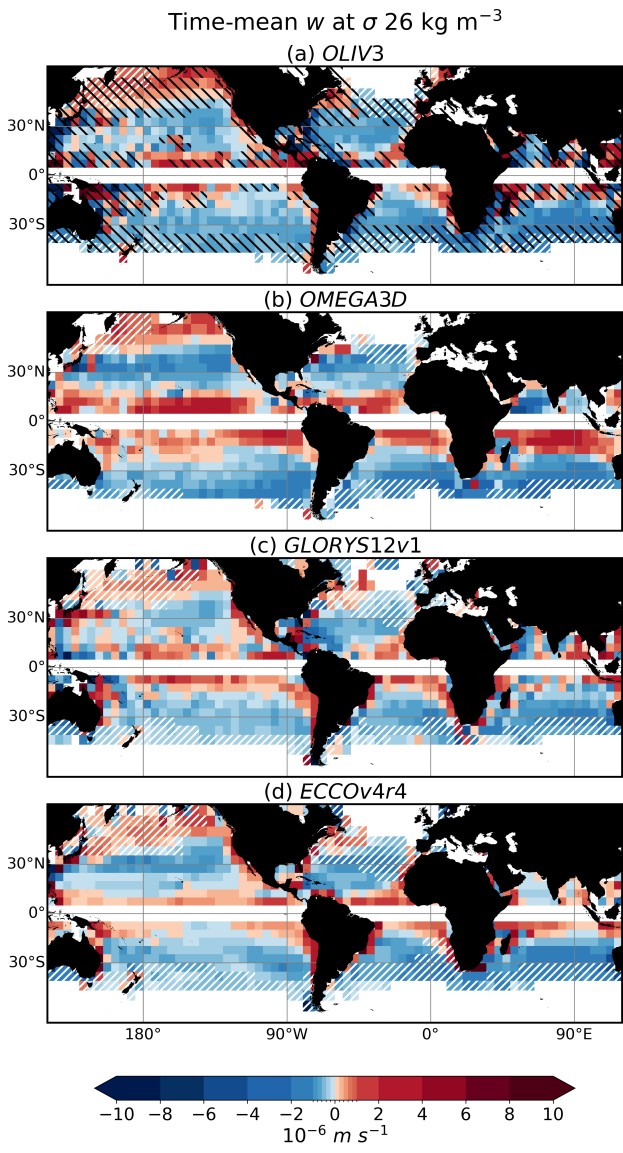

**Figure 4.** Time-mean vertical velocity fields at $\sigma 26$ and 5° x 5° spatial resolution from: (a) OLIV3, (b) OMEGA3D, (c) GLORYS12v1 reanalysis, and (d) ECCOv4r4 reanalysis. White hatching represents regions within the maximum mixed layer defined by the thermohaline field corresponding to each velocity. In panel (a), black hatching indicates the areas where the LVB is not satisfied in the OGCM (relative error >10%).

where the relative differences between OGCM's $w_g$ and $w_{tot}$ are typically below 10%, the relative differences between OLIV3 and any of the reanalyses often exceed this threshold. This suggests that much of the observed differences derive from the

observation-based input fields (ARMOR3D meridional velocities and ERA5 wind stress) rather than the geostrophic compo-





nent reconstructed, which mostly dominates the total flow in the subtropics and upper tropics (Fig. 4). Particularly, western boundary current systems correspond to regions with large errors in the geostrophic LVB-derived vertical velocities (hatching in Fig. 4a). In these regions, additional terms of the vorticity equation, such as the bottom pressure torque, close the vorticity budget (e.g. Hughes and De Cuevas, 2001; Gula et al., 2015; Schoonover et al., 2016).

OLIV3 demonstrates a reasonable ability to qualitatively capture the time-mean vertical velocity structure of the major ocean gyres at $\sigma 26$. Even in regions where the LVB assumption is no longer valid, such as the deep tropics, western boundary currents and the subpolar Pacific, OLIV3 estimates often remain within the uncertainty range defined by the intercomparison datasets.

### 3.3.2 Vertical Structure of Vertical Flow

The presence of vertical shear in the vertical velocity field is fundamental for establishing the Sverdrup balance and defining
the volumes of water influenced by these dynamics. In classical Sverdrup theory, vertical velocities in the deep ocean are assumed to be very weak, allowing the vertically integrated meridional transport to be related to the wind stress curl. Global estimates of the vertical velocity offer insights into the fundamental physics of the ocean interior circulation. When the LVB holds, geostrophic vertical velocities in the ocean interior can be interpreted as the residue of the evacuation by meridional transport of the vertical mass flow input from the layer above. In this context, the vertical profile of the vertical velocity field
adds information about the flow evacuation ratio under Sverdrup's framework. As demonstrated in Cortés-Morales and Lazar (2024), this assumption holds reasonably well in subtropical basins but breaks down at high latitudes. This finding is consistent with recent studies that have directly evaluated Sverdrup balance (e.g. Thomas et al., 2014). Despite its relevance, studies focusing on the Sverdrup balance rarely address the vertical structure of the vertical velocity field.

    To further evaluate the ability of OLIV3 to reproduce the vertical structure of the vertical flow, the gradient of the absolute
value of vertical velocity (Eq. 4) for multiple estimates of $w$ is computed (Fig. 5). Most datasets (OLIV3, ECCOv4r4 and GLORYS12v1) feature a reduction in downwelling amplitude with depth across the subtropical gyres, consistent with a baroclinic structure (panels a, c and d in Fig. 5). The largest negative gradients reach values around $-10 \times 10^{-7}$ m s$^{-1}$ / kg m$^{-3}$ in the Atlantic and Indian subtropical gyres, and near $-5 \times 10^{-7}$ m s$^{-1}$ / kg m$^{-3}$ in the Pacific Ocean. Compared with the rest of estimates, OMEGA3D showcase a more barotropic profile (Fig. 5b). OLIV3 (Fig. 5a) generally captures the reanalyses'
downwelling weakening with depth in the subtropics, but it displays positive vertical gradients in the eastern tropical gyres, indicating increasing magnitude with depth, which contrasts with the other datasets. This difference may arise from a shallower thermocline at tropical latitudes compared to the subtropics (Salmon, 1982), which causes $\sigma 27$ to lie below the bottom threshold of the thermocline. When the gradient is recalculated using $\sigma 26.5$, as a lower limit, OLIV3 captures a negative gradient in both Pacific and Atlantic tropical gyres (Fig. A1 in Appendix A). As shown for the North Atlantic Ocean application of
LVB in Cortés-Morales and Lazar, 2024, tropical gyre's vertical velocities decrease rapidly in the thermocline, remaining an order of magnitude smaller than those at the top of the thermocline. This implies that the lower bounds of the gradient do not substantially bias the vertical structure in a model simulation, as seen in the small relative gradient error between OGCM $w_{tot}$ and $w_g$ in the Pacific and Atlantic tropical gyres in Fig. 3c. However, there are large uncertainties for observation-based datasets like OLIV3 compared to the reanalyses below the thermocline. OLIV3 shows growing vertical velocities with depth



in regions where the time-mean vertical velocities at a given depth also differ, such as the western boundaries and the deep
tropics. This suggests that when OLIV3 fails to capture the correct amplitude of the vertical flow, it also fails to reproduce the
local vertical structure.

    The comparison with existing estimates demonstrates that OLIV3 reproduces the structure in the subtropics and upper
tropics (particularly above $\sigma26.5$), capturing both the amplitude and vertical structure of the vertical flow, indicating that the
geostrophic component is primary contributor to the observation-based vertical flow and that Ekman pumping vertical velocity
is a suitable boundary condition, consistent with the results of the perfect model test, shown in Fig. 3 and Cortés-Morales and
Lazar (2024).

### 3.3.3   Vertical Velocity Time Variability

The perfect model test (Fig. 3) emphasises the high accuracy in terms of temporal variability of the total vertical velocity by
the geostrophic component. To further assess this accuracy, we evaluate the annual variance and the correlation coefficient (R)
across the various intercomparison datasets.

    The key role of mesoscale activity (Wunsch, 2007) can be detected in the annual variance of vertical velocity at an isopycnal
level within the tachocline (Fig. 6). In OLIV3, GLORYS12v1 and ECCOv4r4 (Fig. 6a, c and d), the highest variance values
are found along western boundary current regions and in the lower-tropical band, while the subtropical gyre interior below the
mixed layer generally displays lower variance. This pattern is consistent with known regions of high mesoscale eddy activity,
such as the western boundary current systems and the deep tropics (e.g. Wunsch, 2007; Barceló-Llull et al., 2025), that is
transported into the ocean interior. OMEGA3D (Fig. 5b) deviates from this behaviour, showing a poleward variance increase
near the intersection of the isopycnal level with the ocean surface. ECCOv4r4 (Fig. 6d) maintains a similar variance spatial
distribution compared with OLIV3 and GLORYS12v1 but with a weaker variance gradient between the subtropical gyre centres
and the western boundary current regions, due to lower maximum values. Variance values within the tropical Indian basin, as
well as the western tropical Pacific and Atlantic basins, exhibit considerable uncertainty across datasets. Nevertheless, OLIV3
reconstructs a field with variance comparable to that in GLORYS12v1, even in regions where LVB does not hold. This supports
the ability of the geostrophic component to capture the temporal variability of vertical motion at first order, as evidenced by
Fig. 3.

In addition to the variance, assessing the ability of OLIV3 to represent interannual variability of $w$ is evaluated through the
correlation coefficient between dataset pairs at low resolution at the representative $\sigma26$ (Fig. 7) and as a function of latitude
at three different sigma surfaces (Fig. 8). Across most dataset pairs, the highest correlation values are found in the centres of
the subtropical gyres, while the lowest occur in the tropical band and western boundary currents. The comparison of the two
reanalyses (Fig. 7a and Fig. 8) showcases an overall relatively low correlation over large fractions of the global thermocline,
with maxima within subtropical gyres and parts of the deep tropics, but with latitudinal median values remaining below 0.5
almost everywhere. The comparison of these two reference datasets provides reference values quantifying the uncertainties
inherent to the estimation of the vertical velocity component of the flow.



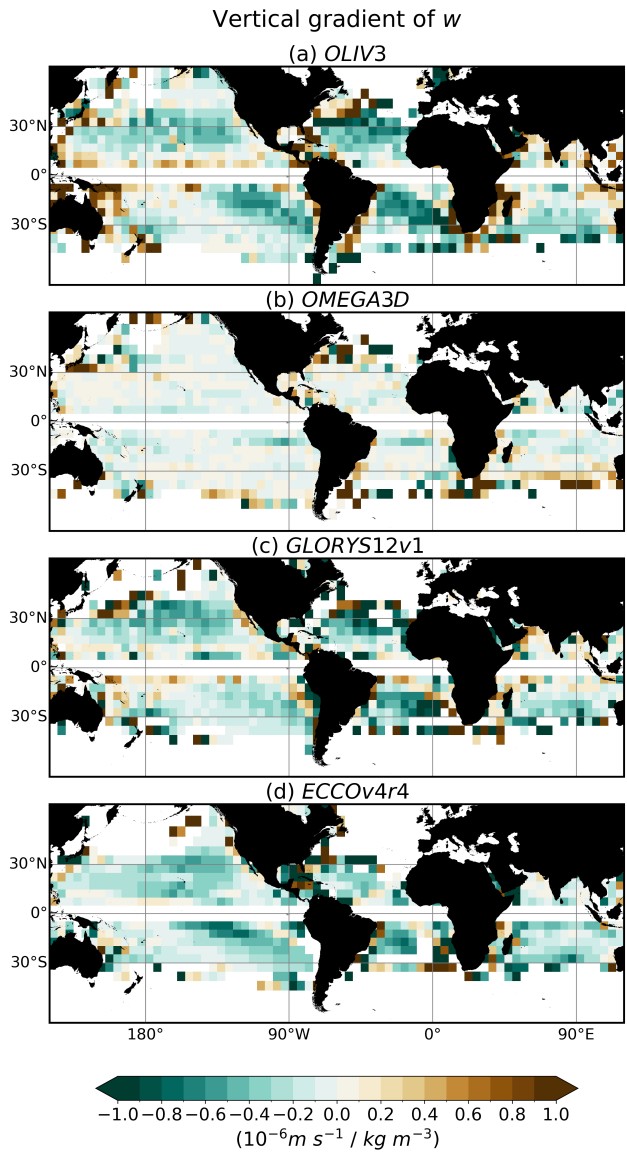

**Figure 5.** Vertical gradient of time-mean vertical velocity between the base of the maximum mixed layer depth and $\sigma 27$ (Eq. 4) at 5° x 5° spatial resolution, shown for: (a) OLIV3, (b) OMEGA3D, (c) GLORYS12v1, and (d) ECCOv4r4. Negative values indicate a decrease in vertical velocity magnitude with depth, while positive values depict increasing magnitude with depth.

It appears that OLIV3 exhibit significant large correlations (R > 0.6) with the reanalyses across large portions of the global subtropics, with values exceeding 0.8 in the Pacific and Atlantic Oceans (Fig. 7c and 7e). In tropical regions where the perfect model test indicates weak correlation between $w_g$ and $w_{tot}$ (hatching in Fig. 7a), R typically falls below 0.4. Notably, reanalyses generally show a lower correlation with each other than with OLIV3 in most of the global subtropical band (Fig. 7a vs. 7c,


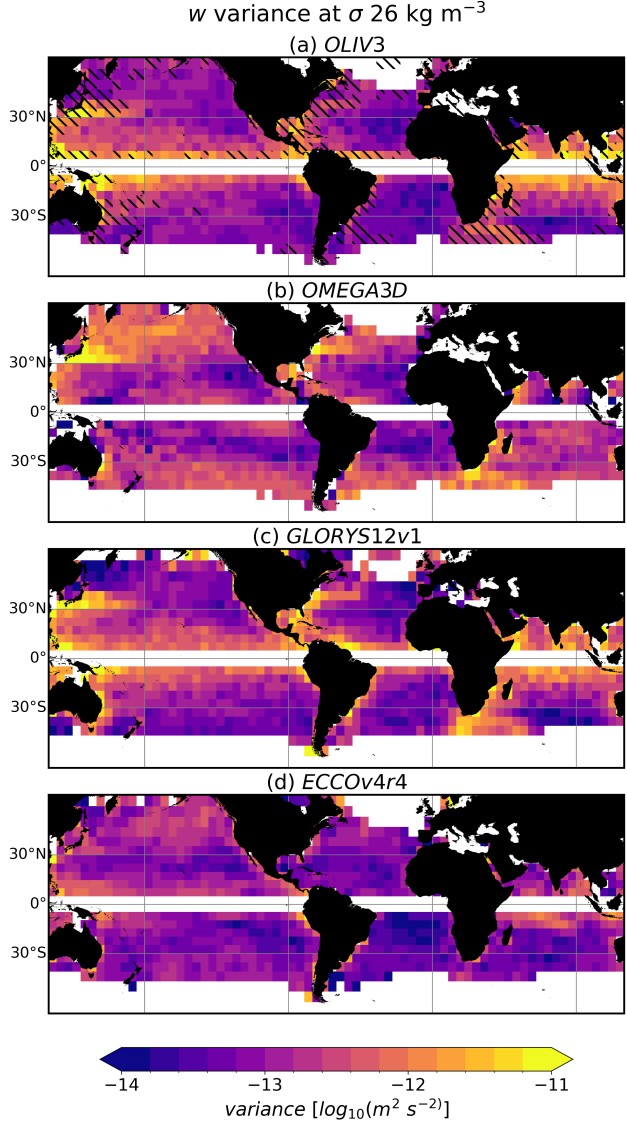

**Figure 6.** Annual variance (logarithmic scale) of vertical velocity at $\sigma 26$ and $5° \times 5°$ spatial resolution for: (a) OLIV3, (b) OMEGA3D, (c) GLORYS12v1, and (d) ECCOv4r4. In panel (a), black hatching represents regions where correlation coefficient between OGCM $w_{tot}$ and OGCM $w_g$ is smaller than 0.5 (Fig. 3b).

7e). As shown in Fig. 3, the geostrophic component dominates the interannual variability of the vertical flow at these depths. Therefore, the reduced inter-reanalysis correlation is likely evidence of the lack of synchronisation in the nonlinear components of vertical flow in assimilated products, while the geostrophic component variability, captured by OLIV3, remains highly correlated. The magnitude and structure of the variability reproduced by the OLIV3 fall within the envelope of variability spanned by very commonly used reanalysis-based estimates of $w$. Although OMEGA3D reaches significant correlation values



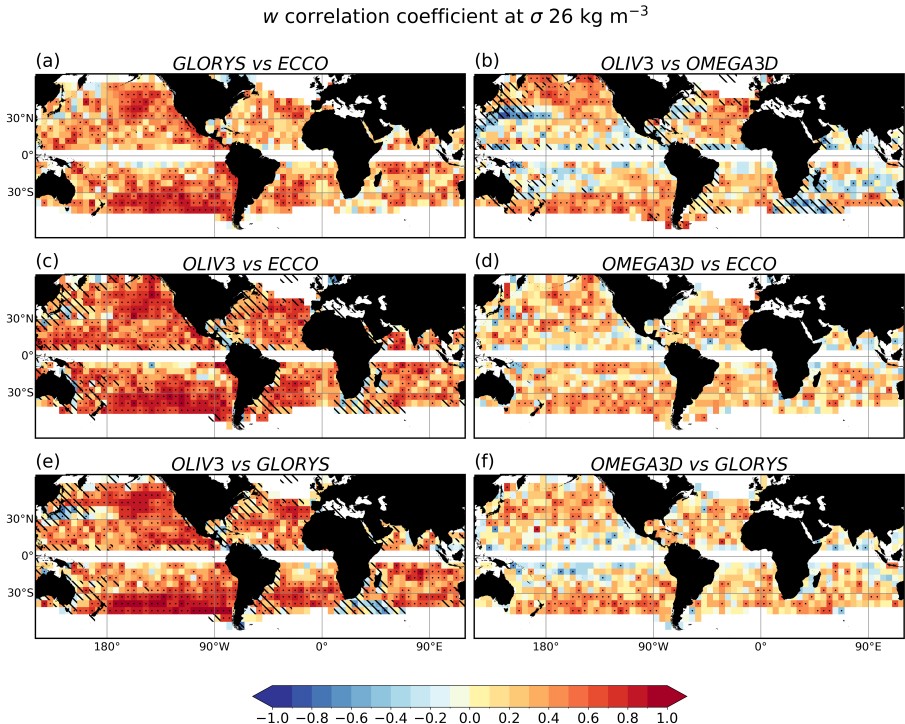

**Figure 7.** Correlation coefficient (R) between the vertical estimates from OLIV3,, GLORYS12v1, ECCOv4r4 and OMEGA3D at $\sigma 26$ and $5° \times 5°$ resolution. Dotted squares indicate correlations significant at the 95% confidence level based on the Student t-test. Black hatching represents regions where correlation coefficient between OGCM $w_{tot}$ and OGCM $w_g$ is smaller than 0.5 (Fig. 3b).

(R up to 0.7) with the intercomparison reanalyses and OLIV3 in some areas within the open ocean, particularly within the Pacific and Atlantic subtropical gyres (Fig. 7b, d and f), its overall performance is poorer and more spatially limited compared to the results for OLIV3.

The general better performance of OLIV3 compared to OMEGA3D in capturing the temporal variability is further illustrated in Fig. 8, which displays the median correlation coefficient value as a function of latitude at three isopycnal levels. Latitudinal median correlation values between OLIV3 and ECCOv4r4 in the Northern Hemisphere subtropical band (20-40°N) are reduced from around 0.6 at $\sigma 25.5$ to 0.4 at $\sigma 27$. Although correlation coefficients between datasets tend to weaken with depth, OLIV3 (solid lines) consistently exhibits higher correlations with the model and reanalyses than OMEGA3D (dashed lines) throughout

the entire thermocline except for high latitudes in the Northern Hemisphere.

### 3.4    Unveiling Methodological Differences Among the Existing Estimates

Several fundamental methodological differences among OLIV3, OMEGA3D and the two reanalyses may account for the discrepancies observed in the climatological horizontal and baroclinic structure, as well as the temporal evolution of the vertical



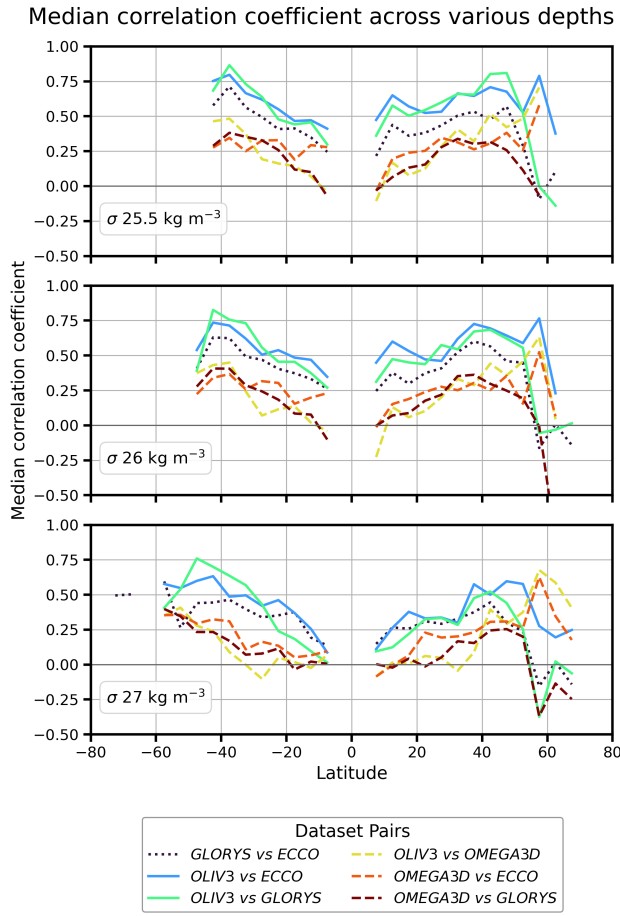

**Figure 8.** Median correlation coefficient (R) as a function of latitude for $\sigma 25.5$, $\sigma 26$ $\sigma 27$. The analysis includes only regions where the correlation between OGCM $w_g$ and $w_{tot}$ exceeds 0.5.Solid, dashed and dotted lines correspond to OLIV3, OMEGA3D intercomparisons, and renalyses intercomparison respectively.

movements. These include the reconstruction methodology, the components of vertical velocity reconstructed, the atmospheric

forcing, the three-dimensional horizontal velocity inputs, and the spatiotemporal resolution of the datasets.

  Atmospheric forcing does not appear to be the primary source of discrepancies, as all datasets employ variants of ERA atmospheric reanalysis products to force the oceanic surface. Similarly, both OLIV3 and OMEGA3D are based on the ARMOR3D geostrophic velocity field, while reanalyses compute vertical velocities directly from the total assimilated horizontal velocity field. Despite their common input, OLIV3 and OMEGA3D exhibit large differences. In contrast, OLIV3 aligns more closely

with the reanalyses despite their distinct origin in either observation-based or assimilated horizontal velocity fields. However, as discussed in Cortés-Morales and Lazar (2024), the ageostrophic component of the horizontal velocity field is negligible in



most of the tropical and subtropical tachocline. Again, while these differences may induce some discrepancies, they are not dominant.

Differences in native spatial and temporal resolution may play a significant role, even when data are averaged to a common resolution. Certain phenomena may be retained across scales and not entirely removed (Yeager, 2015). For example, GLO-RYS12v1 has a resolution of $1/12°$, capturing smaller mesoscale features compared to ECCOv4r4 at $1°$ resolution. Higher spatial resolutions allow GLORYS12v1 to preserve events at seasonal and sub-seasonal timescales, which could not be maintained in coarser native resolutions. Previous studies on reanalysis intercomparison (e.g. Balmaseda et al., 2015; Carton et al., 2019) have shown that such differences due to resolution (in particular in temperature and salinity) are more pronounced in the tropics. In particular, Carton et al. (2018) shows how the coarser resolution of ECCO leads to very distinct results from other eddy-permitting datasets. The spatial resolutions in the study mentioned range from $0.25°$ to $1°$. Therefore, the uncertainty observed here between GLORYS12v1 and ECCOv4r4 probably reproduces and magnifies the uncertainty observed in the cited studies.

OLIV3 and reanalyses estimate $w$ through vertical integration of horizontal velocity fields, because the governing equations only contain the vertical derivative of $w$. In contrast, OMEGA3D employs the omega equation, which incorporates the second-order vertical derivatives and horizontal derivatives of $w$. Consequently, Dirichlet (vertical velocities are set to zero) and Neumann (partial derivatives of vertical velocity are set to zero) conditions are imposed as boundary conditions (Buongiorno Nardelli et al., 2018; Buongiorno Nardelli, 2020). The inclusion of the horizontal and second-order derivatives of $w$ in the OMEGA3D framework may explain the discrepancies with the other datasets. However, a comprehensive examination of the sources of these differences would require a separate in-depth investigation, given the complex physics, constraints and assumptions underlying the omega equation.

### 3.5 Near Surface Interannual Variability of Vertical Flow: Improvement Relative to Ekman Pumping

Beyond the importance of providing an estimate of the vertical profile of thermocline vertical velocities, one might wonder how near-surface $w_g$ compares with Ekman pumping ($w_{Ek}$), the most commonly used observation-based reference product for vertical transfers between thermocline and nitracline, and the surface waters. Indeed, this wind-based computation is frequently employed to validate against observations, or calculate water mass fluxes (e.g. Marshall et al., 1993; Lazar et al., 2002), as well as transport of biogeochemical tracers (Oschlies, 2002) and marine ecosystem parameters, from fish (Parrish et al., 1981) up to whales (Croll et al., 2005). In these types of studies, Ekman pumping is generally considered as a vertical velocity proxy at a variety of levels depending on questions, time-scales and community habits. This level ranges from the bottom of the Ekman layer to that of the winter mixed layer (Williams et al., 2006), an isopycnal surface close to $\sigma 26$ or a fixed depth of a few tens of meters up to 200 meters (Palter et al., 2013). Here, we chose to present results in both isopycnal and depth frames, on $\sigma 26$ and at 100 m depth. Again, the computation is conducted with the reference OGCM simulation, considered to be a dynamically coherent estimate of the real ocean. Figure 9 displays the correlation between $w_{Ek}$ and $w_{tot}$ on $\sigma 26$ and the difference with the correlation between $w_g$ and $w_{tot}$ shown Fig. 3b. Panels c and d in Fig. 9 illustrate the the correlation between $w_g$ and $w_{tot}$ and $w_{Ek}$ and $w_{tot}$ at 100m depth, respectively.



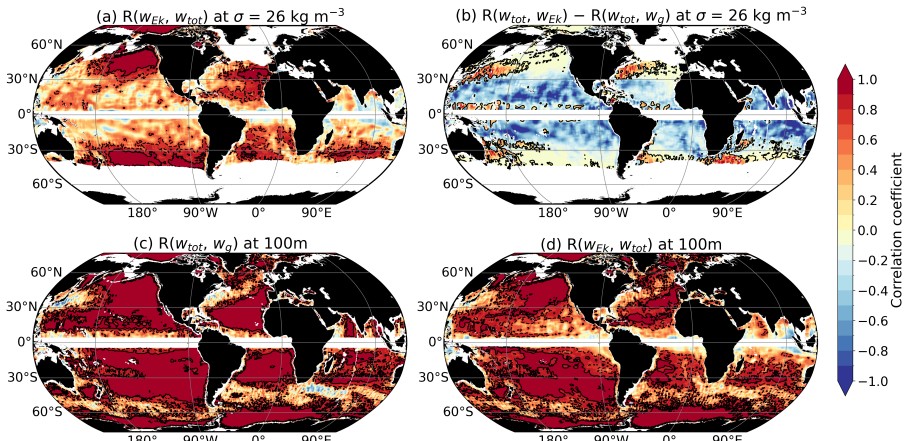

**Figure 9.** (a) Map of correlation coefficients for OGCM $w$ at $\sigma 26$ with Ekman pumping (1993-2015). (b) Differences between Fig. 3b and Fig. 9a. (c) Map of correlation coefficients for OGCM $w$ and $w_g$ at 100 m depth. (d) Map of correlation coefficients for OGCM $w$ at 100 m depth with Ekman pumping. Same isolines as in Fig. 3b for panels a, c and d. For panel c the black contour lines represent the zero. The fields have been smoothed with a $5°$ running mean.

Ekman pumping estimates well (R > 0.7) up to very well (R > 0.9) the variability of total vertical velocities along this isopycnal surface over a relatively small proportion of the thermocline, limited to certain parts of the subtropical gyres and the tropical Atlantic Ocean (Fig. 9a). However, comparison with the correlation between $w_{tot}$ and $w_g$ (Fig. 3b), quantified in Fig. 9b, shows that $w_g$ is accurate over much larger areas of the globe. In other words, $w_{Ek}$ is only more accurate than $w_g$ in

regions where both estimates are poor (R < 0.7), such as western boundary currents and intense open ocean currents, such as the NAD and ACC with a strong zonal component. A marked decrease in the correlation coefficient is observed towards the western boundaries, where the isopycnal surfaces deepen, within the western boundary current systems, as well as in the deep tropics. When comparing this pattern to the correlation between $w_g$ and $w_{tot}$ shown in Fig. 3b, it is remarkable that the areas with high correlation are even more extensive in all basins.

We extended the comparison to other vertical levels, in particular to depths of 50 (not shown) and 100 m (Fig. 9c and d), and reached the same conclusion. Overall, these results show that the Ekman pumping model alone is insufficient to account for interannual variability in vertical flux in most regions of the globe. However, the inclusion of meridional transport divergence in Ekman pumping, i.e. $w_g$, brings about a significant improvement. This highlights the advancement of geostrophic vertical velocities as an estimator of total vertical flux. Moreover, vertical movement in most eastern boundary upwelling systems, such

as Benguela or Peru, shows weak correlations with Ekman pumping. This is consistent with the importance of remote forcing from trapped waves on the coast in these regions (Polo et al., 2008; Illig et al., 2014; Bachèlery et al., 2016).





## 4   Conclusions

A novel observation-based global dataset of geostrophic vertical velocities ($w$) within the thermocline (Observation-based Linear Vorticity Vertical Velocities; OLIV3) is presented and validated against other existing estimates. It is derived from the depth-integrated geostrophic linear vorticity balance (LVB) applied to in situ and satellite geostrophic meridional velocity fields and satellite surface wind stress. OLIV3 provides global ocean estimates of geostrophic vertical velocities at horizontal $0.25°$ resolution, 71 vertical levels and annual frequency, covering the period 1993 - 2019. The product estimates, to the first order, significant parts of the main well-known features of the large-scale tropical and subtropical global tachocline vertical circulation at effective horizontal scales larger than $5°$, in agreement with previous estimates. Most importantly, it also allows a reconstruction of the interannual variability of most of the ocean thermocline, including eastern basin boundaries, to the notable exception of western boundaries, the North Atlantic Drift (NAD) and the Antarctic Circumpolar Current (ACC).

The feasibility of the LVB framework for reconstructing vertical flow is demonstrated by the good agreement between the OGCM geostrophic vertical velocity estimates and the model's total velocity output. The spatial distribution and vertical shear of the temporal means of total velocity are well-reproduced by the geostrophic estimates, except in the deep tropics and along western boundaries across all basins, where nonlinear components of $w$ have a substantial contribution. Furthermore, despite some mean biases, the model's geostrophic vertical velocity field effectively captures the interannual variability of vertical flow, with correlations exceeding 0.9 across most of the tropical, subtropical and extratropical global oceans, excluding western boundary currents and also relatively intense zonal currents of the deep tropics, the NAD and the ACC. This analysis confirms the dominance of geostrophic meridional transport in driving the interannual variability, while nonlinear components primarily influence the time-mean amplitude.

A comparison of OLIV3 against three independent datasets, including two reanalyses (ECCOv4r4 and GLORYS12v1) and an observation-based product (OMEGA3D), over the 1993-2015 period, demonstrates that OLIV3 reproduces the large-scale horizontal patterns and baroclinic vertical structure of the climatological tachocline circulation. It also captures the interannual variability in most of the open-ocean tropical and subtropical regions when compared to the reanalyses. The poorest performance of OLIV3 in the various metrics analysed is found across the western boundary currents, the zonal tropical currents, the NAD and the ACC, aligning with the OGCM preliminary test. Remarkably, OLIV3 often correlates as well or better with each reanalysis than the reanalyses do with each other, suggesting that the geostrophic signal variability is coherently captured, while nonlinear components suffer from a high degree of uncertainty in present estimates of global ocean thermocline vertical velocities.

With the introduction of OLIV3, two observation-based three-dimensional estimates of the vertical velocity field are now available: OLIV3 derived from the LVB, and OMEGA3D based on the quasi-geostrophic omega equation. The intercomparison with OMEGA3D evidences the systematic improvement offered by OLIV3. In particular, OMEGA3D reproduces a relatively barotropic structure, which contrasts with the vertical shear observed in other products and the baroclinic ocean required to sustain the Sverdrup balance. Additionally, OMEGA3D exhibits an overall lower interannual synchrony with reanalyses. We suppose that the discrepancies using the omega equation compared to the LVB arise from the large number of terms in the

former, including higher-order vertical and horizontal derivatives of $w$, many of which require boundary conditions extremely difficult to compute.

An additional comparison within the OGCM reveals that geostrophic vertical velocities ($w_g$) and Ekman pumping ($w_{Ek}$) are strongly correlated across large portions of the global tropical and subtropical gyres. Nonetheless, this coupling weakens
toward the western boundaries and the deep tropics, indicating that meridional transport dynamics, from which the geostrophic component of the vertical flow is derived, increasingly dominate the variability of the vertical flow. Furthermore, the analysis demonstrates that $w_g$ offers a systematically better or equal accuracy in capturing the interannual variaiblity of the total vertical flow than $w_{Ek}$, except in regions of relatively intense zonal currents like the ACC, the tropical zonal currents and countercurrents, and the NAD, where both estimates exhibit limited confidence. We propose to extend this result to the real ocean and
consider that OLIV3 should be more accurate than Ekman pumping for estimating mass or tracer fluxes between the surface and the pycnocline and nutrientcline layers.

OLIV3 is a new and useful tool for investigating interannual variability in polar, subtropical, and tropical gyres, with the exception of their western boundary currents, as well as most parts of the eastern boundary upwelling systems, offshore of the continental plateau. We strongly encourage its use in biogeochemical and biological studies focused on the vertical exchange
of ocean biogeochemical tracers and their impact on marine ecosystems.

Overcoming the current limitations in OLIV3 (spatiotemporal scales coarser than 5° and one year) would require incorporating full meridional velocities and additional terms from the vorticity equation, such as the horizontal advection of relative vorticity. This improvement could extend OLIV3's applicability to finer spatial and temporal scales, including seasonal and monthly variability, and to coastal regions while maintaining the simplicity of the depth-integrated formalism. Detailed works
focusing on regions strongly affected by climate change and extreme events will be conducted in future studies using the upcoming improved version of OLIV3.

*Code and data availability.* The OLIV3 dataset developed in this study is available on 50 vertical levels (https://doi.org/10.5281/zenodo.16981061; Cortés-Morales and Lazar, 2025a) and 71 isopycnal levels (https://doi.org/10.5281/zenodo.16962780; Cortés-Morales and Lazar, 2025b). A low-resolution version, used in the intercomparison test of this study, is available upon request.
Code in MatLab and Python to compute geostrophic velocities (OLIV3 and OGCM), apply the linear vorticity balance and calculate the intercomparison metrics is available at the following repository: https://github.com/dcortales/compute_wglvb.

## Appendix A: Supplementary figures

*Author contributions.* DCM and AL conceptualized and designed the study. DCM processed the data, produced the figures and first draft of the manuscript, together with the associated data products. All authors have reviewed the manuscript. All authors have read and agreed to
545 the published version of the manuscript.



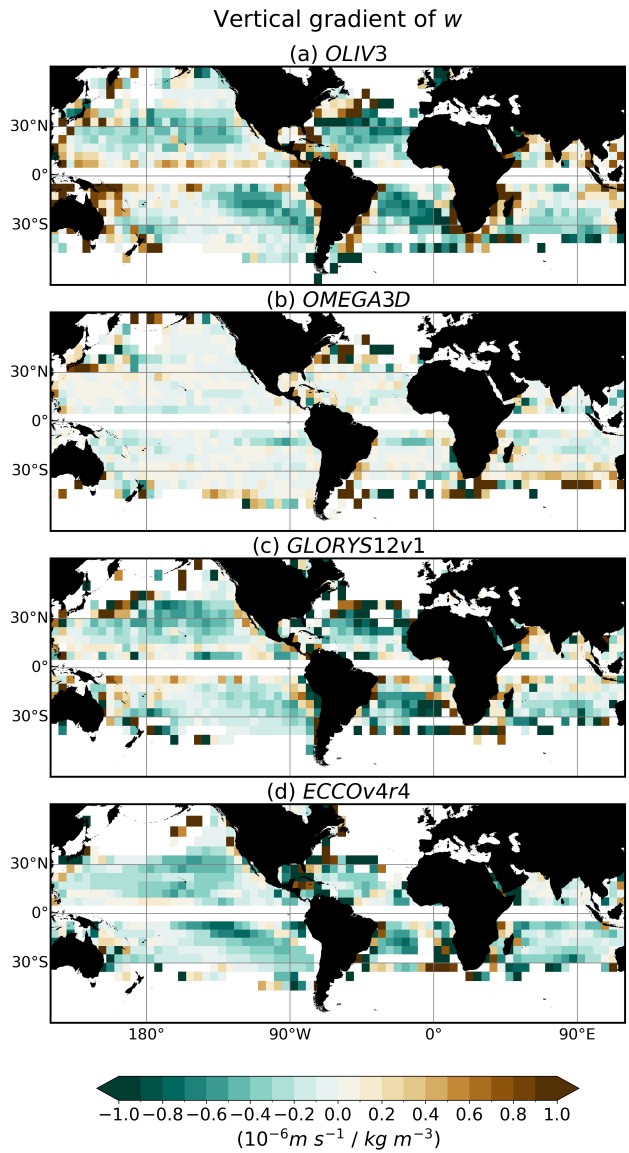

**Figure A1.** Vertical gradient of time-mean vertical velocity between the base of the maximum mixed layer depth and $\sigma26.5$ (Eq. 4) at $5° \times 5°$ spatial resolution, shown for: (a) OLIV3, (b) OMEGA3D, (c) GLORYS12v1, and (d) ECCOv4r4. Negative values indicate a decrease in vertical velocity magnitude with depth, while positive values depict increasing magnitude with depth.

*Competing interests.* The contact author has declared that none of the authors has any competing interests.





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
