# Peer review of "Global Thermocline Vertical Velocities: a Novel Observation Based Estimate"

_Earth System Science Data, 2025_

## Referee Comment (RC1)

**Review of the paper entitled Global Thermocline Vertical Velocities: a Novel Observation Based Estimate**

**1 Major Points**

This study proposes the global OLIV3 product of geostrophic vertical velocity in the thermocline, which is derived from ARMOR3D observation-based meridional geostrophic currents and ERA5 surface wind-stress used to derive Ekman pumping. The methodology adopted in OLIV3 relies on the linearization of the vorticity equation, where the vertical stretching term balances the meridional advection of planetary vorticity. The validations of OLIV3 against the perfect OGCM model and the GLORYS12v1 and EC-COv4r4 reanalyses are convincing. OLIV3 captures the interannual variability of tropical and subtropical regions, but fails at fronts and boundary current systems, precisely where subduction and modal water formtion occur. Regions between mesoscale structures are also populated by submesoscale structures such as filaments, eddies, whose contribution in terms of integrated vertical transport represents about 50% of the total vertical transport (Klein et al., XX ? I don't remember the date). In short, OLIV3 is efficient in large-scale structures, but has significant shortcomings in crucial regions.

Many products, such as reanalyses, and OGCM, OAGCM models outputs, produce vertical velocities in the thermocline, even at high frequencies. It is therefore important to demonstrate the added value of the OLIV3 database in comparison with these models and reanalyses, simply because models and reanalyses provide the full vertical velocity, which is an important variable for biogeochemistry, for example. Deriving the vertical velocity from the complete vorticity equation or the omega equation shed light into processes driving vertical motion, as well as the balances between these processes. Here the added values of these approaches. Your article should demonstrate the usefulness and applications of the OLIV3 database, and not just show that the main large-scale balance lies between meridional advection of the planetary vorticity and vertical w-stretching.

The meridional geostrophic velocity  $(v_g)$  is taken into account in the linear balance equation. I don't understand why the geostrophic vorticity was not conserved in the balance equation  $(f + \zeta_g) \frac{\partial w}{\partial z} = \beta v_g$ ?

I do not understand how the Ekman pumping is taken into account. In fact, I suspect it is  $w_{tot} = w_g + w_{ek}$ . From my understanding it is about:

$$f \int_{h}^{0} \frac{\partial w}{\partial z} dz \simeq f \int_{h}^{z_{geo}} \frac{\partial w}{\partial z} dz + f \int_{z_{geo}}^{z_{ek}} \frac{\partial w}{\partial z} dz = \beta \int_{h}^{0} v_{g} dz \tag{1}$$

where h,  $z_{geo}$  and  $z_{ek}$  are the level of no motion, the depth of thermocline where  $w_g$  is computed and the depth of the Ekman layer, respectively.

$$\begin{cases}
f(w_g + w_{ek}) = fw_{tot} = \beta \overline{v}_g h \\
\overline{v}_g = \frac{1}{h} \int_h^0 v_g dz
\end{cases}$$
(2)

This point is essential and must be clarified.

It is mentioned in the conclusion that total meridional velocities  $(v_g + v_{ag})$  and additional terms from the vorticity equation, such as the horizontal advection of relative vorticity should be incorporated. In some way, you have already incorporated an ageostrophic component of the current with Ekman pumping. By introducing  $v_{ag}$  in equation  $f\frac{\partial w}{\partial z} = \beta(v_g + v_{ag})$ , what do you expect on OLIV3 performances?

How do you intend to compute  $v_{aq}$ ?

To have consistency between  $v_g$  and  $v_{ag}$ , isn't it better to use v and also the vorticity  $\zeta$  from a reanalysis?

Figures 4, 5, and 6 show OLIV3, GLORYS12, ECCOV4 and OMEGA3D. However, it is the differences between OLIV3 and these other products that are discussed. These differences are very difficult to see. Please provide Figures illustrating these differences.

It is better to use the vertical velocity in m/day rather than in m/s.

**2 Detailed Points**

- Line 76: typos: change "gesotrophic" to "geostrophic"
- Line 98: "local mass balance between meridional divergent flow and an opposing vertical convergence". Ok but what equation 1 shows is a balance between vertical convergence and meridional advection of planetary vorticity.
- Line 101: This sentence is confusing because the horizontal geostrophic flow is non-divergent. Please reformulate.
- Line 106-107: The Ekman pumping  $w_{ek}$  occurs at the Ekman pumping depth ( $D_{ek} = 0.2*\sqrt{\tau}/f$ , Li et al., 2021; GRL). So a vertical profile of  $w_{ek}$  is often prescribed from the surface, where w = 0, to  $z = z_{ek}$ , where  $w = w_{ek}$ , to  $z = 2D_{ek}$ , where w = 0. So  $w = w_{ek}$  at z = 0 is not a good surface boundary condition. Please correct.
- Line 133: Ekman pumping is not clearly shown in Equation 2. See remark in the "Major Points" Section.
- Line 135: Here again it is not  $w_g$  because it includes  $w_{ek}$ . This is confusing because, as said lines 131-132, the product w of OLIV3 has two components, which are  $w_g$  and  $w_{ek}$ . Please clarify this point.
- Line 150: Isn' it better to calculate  $v_g$  from the thermal wind equation? Based on pressure, the result is often noisy, unless the pressure is first smoothed. In this case, the filtering procedure should be mentioned.

- Line 153: The reference Jean-Michel et al., 2021 is not adequate. You cannot use the first name of the authors in references. Please correct.
- Line 165: Omega-equations need not only surface momentum and heat air-sea fluxes, but also fluxes in the ocean. Where do these fluxes come from ?
- Line 178: The equator band (5S/N) is large. Geostrophism can be applied from 2S/N degrees, and even 1S/N degree. For example see Dourado and Caniaux, JGR, 2001 (their Figure 4).
- Line 182-184: Why was the isopycnal level  $\sigma$ 26 chosen? How does it compare to the the mixed-layer depth? Why not choose the mixed-layer depth?
- Line 188-189: Explain why Figure 1 emphasises the role of atmospheric forcing as the primary driver of vertical flow within the upper ocean. Are you implying that the Sverdrup balance can be used to obtain a good estimator of  $v_q$ ?
- Line 194-195: This aspect is an issue because we do not see the point of using LVB. Please identify and discuss the missing processes in the LVB to correctly represent the frontal dynamics.
- Line 196-197: Please show a Figure of Ekman pumping.
- Line 230: Equation 4. Using  $\sigma 27 \sigma_{MLD}$  makes difficult to understand the following discussion, because we do not know the sign of this difference. Then the speech is difficult to follow. I suggest the metric  $\left(\frac{\partial w_g}{\partial z}\right)_{z=MLD} \left(\frac{\partial w_{tot}}{\partial z}\right)_{z=MLD}$  instead, normalized or not.
- Line 245-250: This shows the limits of the method in frontal regions. Even in coastal regions, Ekman pumping fails to capture vertical transport of physical and biogeochemical tracers.
- Line 272-273: I don't understand why a downward decrease of  $w_g$ . I would instead expect a positive vertical gradient. I am having trouble following the discussion about the vertical gradient of w, because the sign of  $(\sigma 27 \sigma_{MLD})$  is unclear.
- Line 330-331: Reanalyses are significantly affected by spin-up effects, primarily vertical velocity. This is why incremental analysis update techniques are used in data assimilation procedures. Consequently, how much confidence can we place in such reanalysed vertical velocities, given that they are partially affected by unphysical spurious effects? In other words is it reasonable to use them as w-references?
- Line 335: Reanalyses are significantly affected by spin-up effects.
- Line 348-349: I don't understand this sentence. I would say that the geostrophic vertical velocity in the ocean interior results from the convergence/divergence of the Ekman drift.

- Line 355: Figure 5. Sorry but I am lost with vertical gradient expressed in  $ms^{-1}/kgm^{-3}$ . Where does  $\sigma 27$  fit in relation to  $\sigma_{MLD}$ ? I suggest expressing this gradient in  $day^{-1} = mday^{-1}/m$ .
- Line 375-376: Be careful  $w(z=0) \neq w_{ek}$ .
- Line 387: Change Fig.5b to Fig.6b.
- Line 391-393: Arbitrary conclusion at first glance (Fig 6). Make difference maps.
- Line 428: Not good due to spin-up.
- Line 446: OMEGA3D also integrates vertical stratification.
- Line 450: OMEGA3D is a physical investigating tool because it is based on the destruction of the thermal wind balance by current and turbulence.
- Line 527-530: If I am a biogeochemical scientist, or physicist who wants to estimate modal water production, what is the benefit of using OLIV3 rather than a reanalysis? Sorry, I'm not convinced, but I would like to be.
- Line 532: Before incorporating non-linear processes, integrate before the total meridional velocity and vorticity.

In conclusion, I request substantial changes, particularly on the interest of using OLIV3, and clarifications on the incorporation of Ekman pumping in Equation 2, and the physical interpretation of this equation balance.

---

## Author Comment (AC1)

**Review of the paper entitled Global Thermocline Vertical Velocities: a Novel Observation Based Estimate**

**1 Major Points**

This study proposes the global OLIV3 product of geostrophic vertical velocity in the thermocline, which is derived from ARMOR3D observation-based meridional geostrophic currents and ERA5 surface wind-stress used to derive Ekman pumping. The methodology adopted in OLIV3 relies on the linearization of the vorticity equation, where the vertical stretching term balances the meridional advection of planetary vorticity. The validations of OLIV3 against the perfect OGCM model and the GLORYS12v1 and ECCOv4r4 reanalyses are convincing. OLIV3 captures the interannual variability of tropical and subtropical regions, but fails at fronts and boundary current systems, precisely where subduction and modal water formation occur. Regions between mesoscale structures are also populated by submesoscale structures such as filaments, eddies, whose contribution in terms of integrated vertical transport represents about 50% of the total vertical transport (Klein et al., XX ? I don't remember the date). In short, OLIV3 is efficient in large-scale structures, but has significant shortcomings in crucial regions.

Many products, such as reanalyses, and OGCM, OAGCM models outputs, produce vertical velocities in the thermocline, even at high frequencies. It is therefore important to demonstrate the added value of the OLIV3 database in comparison with these models and reanalyses, simply because models and reanalyses provide the full vertical velocity, which is an important variable for biogeochemistry, for example. Deriving the vertical velocity from the complete vorticity equation or the omega equation shed light into processes driving vertical motion, as well as the balances between these processes. Here the added values of these approaches. Your article should demonstrate the usefulness and applications of the OLIV3 database and not just show that the main large-scale balance lies between meridional advection of the planetary vorticity and vertical w-stretching.

- Thank you for your remark. The primary objective of OLIV3 is to provide an observation-based and dynamically consistent reconstruction of the large-scale geostrophic vertical velocity field, intended to complement existing products derived from OGCMs, reanalysis and observations. Vertical velocity datasets available for the scientific community often differ substantially in their representation of the large-scale circulation as they rely on different variables and data sources, including models, reanalyses, and observations. In particular, we find a substantial discrepancy in the vertical structure of the vertical velocity between OMEGA3D (an observation-based product derived from the omega equation) and two different reanalyses. Even when datasets originate from similar sources, differences in model configuration, spin-up, parameterizations, assimilation methodologies or the equation used to retrieve the vertical velocities can lead to pronounced different estimates of the vertical flow. OLIV3 offers a dataset based on a robust framework (the Linear Vorticity Balance (LVB)) applied to observationbased variables representing the first-order signal for mean states in many regions and for interannual variability globally. This objective precedes the focus on sub- and mesoscale structures, although we fully acknowledge that these processes explain most of vertical kinetic energy in the upper ocean, as demonstrated by Klein et al., 2008.

A key added value of OLIV3 is that it produces a physically consistent large-scale three-dimensional vertical velocity field that reproduces a more realistic baroclinic structure within the thermocline compared with OMEGA3D required to maintain the Sverdrup balance. As a result, OLIV3 should enable the representation of vertical fluxes of passive traces. This aspect is not reproduced by observation-based fields derived from the omega equation available for the community as represented in Fig. 6.

As discussed in Cortés-Morales and Lazar (2024), the LVB framework has indeed notable limitations in regions where the dominant dynamics are sub- and mesoscale, frontal, or strongly ageostrophic (e.g. boundary currents). These limitations are inherent to the scales where the balance holds, which are larger than those characteristic to sub- and mesoscale processes such as filaments, eddies and fronts. Nonetheless, the balance holds well in large-scale regimes where geostrophic dynamics dominate the flow, including key regions such as eastern boundary upwelling systems that play a critical role in defining oxygen minimum zones (OMZs).

Since no "ground-truth" for the vertical velocities exists, by expanding the ensemble of independent reconstructions of the vertical velocity field, each derived from different data and methodologies, we can identify robust large-scale features across methods, and where existing products diverge. In this sense, OLIV3 is not intended to replace high-frequency vertical velocity estimates, but to provide a benchmark based on the available observations that is independent of the numerical biases found in primitive equation models.

The meridional geostrophic velocity (vg) is taken into account in the linear balance equation. I don't understand why the geostrophic vorticity was not conserved in the balance equation $(f + \zeta g)\ \partial w\ \partial z = \beta vg$ ?

- Thank you for raising this point. In our formulation, the geostrophic relative vorticity term is neglected based on its order of magnitude. In the large-scale circulation regime of interest, characterised by a small Rossby number ($Ro$ $<<$ 1), the terms involving the relative vorticity are typically several orders of magnitude smaller than the Coriolis parameter dependent terms. As discussed in Cortés-Morales and Lazar (2024), this scale separation justifies neglecting the contribution of other terms from the vorticity equation, as its impact is negligible compared to f- and beta-dependent components. However, we acknowledge that this is a good general starting point, but not for a point-to-point assessment, and they may be regions where this assumption is no longer valid and the LVB breaks. For this reason, it is included in the dataset product an additional flag variable indicating the OGCM

time-mean relative error and temporal correlation between the total vertical velocity and the LVB-derived geostrophic velocity described in lines 178–180: "*The product is quality-flagged based on the time-mean relative error and interannual correlation coefficient between $w_g$ and $w_{tot}$ in the OGCM perfect model test*". This allows users to identify regions where OLIV3 should be interpreted with caution.

I do not understand how the Ekman pumping is taken into account. In fact, I suspect it is wtot = wg + wek. From my understanding it is about:

Eq. 1 from Review

where h, zgeo and zek are the level of no motion, the depth of thermocline where wg is computed and the depth of the Ekman layer, respectively.

Eq. 2 from Review

This point is essential and must be clarified.

- In this study, we focus on the geostrophic component of the vertical velocity (wg). However, because the total vertical velocity must satisfy the kinematic boundary condition (wtot(z=0) = 0), and considering that the only ageostrophic component of the vertical velocity is the Ekman pumping (wek), wg is necessarily balanced by wek at the surface, i.e., wg(z=0) = -wek(z=0).
  Following your comment, we have revised subsection 2.1 in the Methodology and Data section and added a new Appendix A to provide a more detailed description of the Ekman pumping contribution. In particular, we now explicitly justify why the boundary condition is imposed at the surface rather than at the base of the Ekman layer under a beta-plane approximation, understood as beta that the meridional derivative of f exists locally, following in the discussion in Pedloski (1979), where it is theoretically demonstrated that the divergence of the geostrophic horizontal flow is not zero on a beta-plane, and this justifies the assumptions in our approach.

It is mentioned in the conclusion that total meridional velocities (vg + vag) and additional terms from the vorticity equation, such as the horizontal advection of relative vorticity should be incorporated. In some way, you have already incorporated an ageostrophic component of the current with Ekman pumping. By introducing vag in equation f ∂w ∂z = β(vg + vag), what do you expect on OLIV3 performances ?

- We expect that including the ageostrophic meridional velocity in the balance would extend the regions where the LVB framework can describe the vertical flow. Because the thermocline circulation is largely geostrophic, the inclusion of the ageostrophic meridional velocities is not expected to substantially improve the OLIV3 accuracy in these. However, including the total meridional velocity could extend the accuracy of the vertical flow, closer to the intergyre region for example. The largest potential benefit is expected below the thermocline, where the existence of a level of no motion makes the vorticity balance more sensitive to ageostrophic contributions (Cortés-Morales and Lazar, 2024). Although it is true that we have already included the ageostrophic component with Ekman pumping, we have included its

contribution throughout the Ekman layer, but not its contribution at each level within it and how it affects the total vertical velocity. Furthermore, although geostrophic circulation dominates horizontal circulation in the thermocline of tropical and subtropical gyres, this is not the case in regions such as the WBC (Cortés-Morales and Lazar, 2024), where the ageostrophic component has a non-negligible role. Therefore, Ekman pumping is not the only ageostrophic component of vertical circulation.

How do you intend to compute vag ?

- In the ocean interior, the flow is mainly geostrophic. However, it is not the case in the ocean interior. Using total currents at these levels from gridded observation-based products, such as GLOBCURRENTS (https://doi.org/10.48670/mds-00327) or AGESC-Med (https://doi.org/10.1016/j.dib.2023.109804) in the Mediterranean Sea and in-situ measurements such as the ones contained in the Global Ocean-Delayed Mode in-situ Observations of surface and sub-surface ocean currents product (https://doi.org/10.17882/86236) could add additional terms to the geostrophic velocities from OLIV3.

To have consistency between vg and vag, isn't it better to use v and also the vorticity ζ from a reanalysis ?

- Thank you for you suggestion. Using reanalysis for the total meridional velocity and relative vorticity is a valid option. However, for OLIV3 we deliberately chose an observation-based reference (ARMOR3D) available for the scientific community. OLIV3 is intended to be an observation-based product.

Figures 4, 5, and 6 show OLIV3, GLORYS12, ECCOV4 and OMEGA3D. However, it is the differences between OLIV3 and these other products that are discussed. These differences are very difficult to see. Please provide Figures illustrating these differences.

- We thank the reviewer for this comment. In the revised manuscript, we have added new figures that explicitly show the differences between OLIV3 and the three comparison products (GLORYS12, ECCOv4, and OMEGA3D) and among them. These new panels are now included alongside the original fields in the new Figures 5-8, allowing the spatial patterns of agreement and disagreement to be visualized much more clearly.

It is better to use the vertical velocity in m/day rather than in m/s.

- All figures and text unit references have been changed from m/s to m/day.

**2 Detailed Points**

- Line 76 : typos: change "gesotrophic" to "geostrophic"
  - Corrected
- Line 98: "local mass balance between meridional divergent flow and an opposing vertical convergence". Ok but what equation 1 shows is a balance between vertical convergence and meridional advection of planetary vorticity.

- o Changed in line 108-109 by: "*Physically, Eq. 1 expresses the balance between the meridional transport of the planetary vorticity by geostrophic flow and the vortex stretching induced by the vertical motion.*"

- Line 101: This sentence is confusing because the horizontal geostrophic flow is nondivergent. Please reformulate.
  - o Thank you for pointing this out. We have thoroughly revised Section 2.1 and added a new Appendix A to improve clarity regarding the divergence of the geostrophic flow and the role of Ekman pumping. In particular, we now explicitly discuss that while the horizontal geostrophic flow is nondivergent on the f-plane, this is not the case on the β-plane (Pedlosky, 1996).

- Line 106-107: The Ekman pumping wek occurs at the Ekman pumping depth (Dek =0.2∗√τ/f, Li et al., 2021; GRL). So a vertical profile of wek is often prescribed fromthe surface, where w = 0, to z = zek, where w = wek, to z = 2Dek, where w = 0. So w = wek at z = 0 is not a good surface boundary condition. Please correct.
  - o Thank you for the comment. We have updated section 2.1 including a discussion following previous literature about the need to define Ekman pumping at the surface in a beta-plane (where the geostrophic divergence is not zero) instead at the Ekman depth. See also our response to the general comment above.

- Line 133: Ekman pumping is not clearly shown in Equation 2. See remark in the "Major Points" Section.
  - o Changed in the update of section 2.1.

- Line 135: Here again it is not wg because it includes wek. This is confusing because, as said lines 131-132, the product w of OLIV3 has two components, which are wg and wek. Please clarify this point.
  - o We have improved the explanation to avoid misunderstanding.

- Line 150: Isn' it better to calculate vg from the thermal wind equation? Based on pressure, the result is often noisy, unless the pressure is first smoothed. In this case, the filtering procedure should be mentioned.
  - o We agree with the reviewer on the possibility of retrieving the meridional geostrophic velocity from the thermal wind relation. However, this approach requires computing vertical derivative of the velocities, implying the depth integral of the temperature. We preferred to not add more steps to the computation to avoid the propagation of errors.

- Line 153: The reference Jean-Michel et al., 2021 is not adequate. You cannot use the first name of the authors in references. Please correct.
  - o Thank you for noticing. In the referenced publication, the First and Family Names are inverted. We have corrected the citation in line 196 and the references list in line 763.

- Line 165: Omega-equations need not only surface momentum and heat air-sea fluxes, but also fluxes in the ocean. Where do these fluxes come from ?
  - o The reviewer is correct that the omega-equations require not only surface air–sea fluxes but also fluxes within the ocean interior. In the OMEGA3D product, forcing terms are computed from ARMOR3D potential density and

geostrophic velocity fields as well as ERA Interim atmospheric reanalyses (Buongiorno Nardelli, 2020). It is mentioned in line 210: *"... and ERA-Interim (Dee et al., 2011) surface air-sea fluxes."*

- Line 178: The equator band (5S/N) is large. Geostrophism can be applied from 2S/N degrees, and even 1S/N degree. For example see Dourado and Caniaux, JGR, 2001 (their Figure 4).
    - Thank you very much for the suggestion and reference. We understand that in theory the geostrophic approximation can be extended closer to the equator. However, we have selected the equatorial mask (5°S–5°N) to remain consistent with the ARMOR3D product (Mulet et al., 2012). From ARMOR3D product QUID (https://documentation.marine.copernicus.eu/QUID/CMEMS-MOB-QUID-015-012.pdf): *"At the equator, the thermal wind equation is no more valid because the Coriolis parameter f is zero. Therefore, the method is adapted between 7°S and 7°N: For the zonal component, the velocities are estimated with a second order differentiation (Picaut & al, 1989)."*
- Line 182-184: Why was the isopycnal level σ26 chosen? How does it compare to the the mixed-layer depth? Why not choose the mixed-layer depth ?
    - Thank you for this question. The isopycnal level σ26 was chosen as a representative depth within the thermocline, where the LVB approximation is valid. In Cortes-Morales and Lazar (2024), we have demonstrated that the σ26 is a representative example of the thermocline. As you can observe in Figure 7 from the same paper, the LVB approximation holds valid below the MLD, so it could be also possible to show this level with the same qualitative properties. This is clarified in lines 226-228: *"This isopycnal level was chosen to assess the vertical velocity estimates across most of the extension of the global subtropical gyres, while maintaining a focus on thermocline dynamics, where the LVB framework performs best (see CM24 for the North Atlantic Ocean)."*
- Line 188-189: Explain why Figure 1 emphasises the role of atmospheric forcing as the primary driver of vertical flow within the upper ocean. Are you implying that the Sverdrup balance can be used to obtain a good estimator of vg ?
    - Thank you for the comment. In the revised Figure 2 (previous Figure 1), we have included an additional panel (b) showing the Ekman pumping at the ocean surface. This allows a direct comparison between the vertical velocities at the ocean surface (Ekman pumping) and those in the ocean interior (OLIV3 at sigma26). The overall agreement in the large-scale patterns of upwelling and downwelling between the two levels suggests the dominant role of the Ekman pumping to the vertical velocity in the ocean interior compared with the divergence of the horizontal geostrophic flow.
    - Regarding the Sverdrup balance, we do not intend to imply that it provides a direct estimator of the geostrophic meridional velocity. Rather, the relationship is conceptual: the computation of OLIV3 can be interpreted as the indefinite integration of LVB (Sverdrup balance being the definite integration of LVB), such that the geostrophic vertical velocity at a given

depth represents the fraction of the atmospheric pumping that is not evacuated by the horizontal divergence above that depth. When the geostrophic vertical velocity at a given depth is effectively null, the divergence of the above horizontal flow fully evacuated the atmospheric input, therefore assuming the Sverdrup balance describes the ocean dynamics in the location up to the given depth. This is now clarified in lines 390-394:" *When the LVB holds, geostrophic vertical velocities in the ocean interior can be interpreted as the residue of the evacuation by meridional transport of the vertical mass flow input from the layer above. If the geostrophic vertical velocity at a given depth is effectively negligible, the divergence of the horizontal flow fully compensates the wind driven divergence above this level, implying that the Sverdrup balance adequately describes the ocean dynamics down to that depth.*"

- Line 194-195: This aspect is an issue because we do not see the point of using LVB. Please identify and discuss the missing processes in the LVB to correctly represent the frontal dynamics.

  o Thank you for the comment. In the western boundary currents (WBC) and other frontal regions, the LVB is no longer a good approximation of the vorticity balance, as shown in Cortés-Morales and Lazar (2024). Previous studies focused on the Depth-Integrated Vorticity Balance equation have demonstrated that the bottom pressure torque (BPT) effectively balances the barotropic planetary vorticity advection, in the WBC, with the wind forcing being negligible (e.g., Hughes and de Cuevas 2001; Gula et al. 2015; Schoonover et al. 2016) as indicated in lines 380-382. Additionally, nonlinear advection of relative vorticity contributes substantially to closing the vorticity budget, as further supported by OGCM analyses in the North Atlantic (Cortés-Morales, 2024, thesis). However, the LVB performs well in regions such as the eastern boundary upwelling systems (EBUS) as indicated in lines 559-561. It is important to note that the LVB captures large-scale patterns, but it cannot resolve finer-scale dynamics, which are dominated by nonlinear and ageostrophic processes (Cortés-Morales and Lazar, 2024).

- Line 196-197: Please show a Figure of Ekman pumping.

  o Thank you for the suggestion. We have included a panel (b) in Figure 2 (previous Figure 1) showing the Ekman pumping.

- Line 230: Equation 4. Using $\sigma27-\sigma MLD$ makes difficult to understand the following discussion, because we do not know the sign of this difference. Then the speech is difficult to follow. I suggest the metric $\partial wg\ \partial z\ z=MLD\ -\ \partial wtot\ \partial z\ z=MLD$ instead, normalized or not.

  o Thank you for pointing out. We have revised Equation 13 (previous Equation 4) to use the vertical distance between isopycnal surfaces (in meters) rather the difference in density (sigma27-sigmaMLD) and we have change the name to "diapycnal gradient". In this formulation, we consider only the magnitude of the distance, not the sign, focusing on regions where the mixed layer is shallower than sigma 27. In this way, negative values indicate

a decrease in magnitude with depth, while positive values indicate an increase. The revised equation and comments are updated in line 275: "*Negative values indicate a decrease in magnitude with depth, while positive values indicate an increase.*"

- Line 245-250: This shows the limits of the method in frontal regions. Even in coastal regions, Ekman pumping fails to capture vertical transport of physical and biogeochemical tracers.
    - We agree with the reviewer that OLIV3 has substantial limitations in the description of the vertical velocities in frontal and coastal areas. The LVB cannot reconstruct completely the vertical flow in these regions because the geostrophic component is not the dominant contributor to the vertical velocity in these areas. However, large uncertainties between the various references still exist. To help users assess the reliability of OLIV3, we have included an additional flag variable indicating the OGCM time-mean relative error and temporal correlation between the total vertical velocity and the geostrophic vertical velocity computed from the depth-integrated LVB as described in lines 178–179: "*The product is quality-flagged based on the time-mean relative error and interannual correlation coefficient between $w_g$ and $w_{tot}$ in the OGCM perfect model test*". This allows users to identify regions where OLIV3 should be interpreted with caution.
- Line 272-273: I don't understand why a downward decrease of wg. I would instead expect a positive vertical gradient. I am having trouble following the discussion about the vertical gradient of w, because the sign of (σ27 – σMLD) is unclear.
    - Thank you for pointing this out. We recognise the unclear wording. We have changed the sentence in the lines 316-319 to clarify the discussion of the diapycnal gradient: "*Note that the diapycnal gradient of the time-mean total vertical velocity is almost everywhere positive (non-dotted areas), indicating a decrease in the magnitude of the vertical velocity toward the base of the thermocline. This structure is consistent with a baroclinic velocity field, generating a tachocline...*" . This is also included in lines 379-382:" *Particularly, western boundary current systems correspond to regions with large errors in the geostrophic LVB-derived vertical velocities (hatching in Fig. 5a). In these regions, additional terms of the vorticity equation, such as the bottom pressure torque, close the vorticity budget (e.g. Hughes and De Cuevas, 2001; Gula et al., 2015; Schoonover et al., 2016).*"
- Line 330-331: Reanalyses are significantly affected by spin-up effects, primarily vertical velocity. This is why incremental analysis update techniques are used in data assimilation procedures. Consequently, how much confidence can we place in such reanalysed vertical velocities, given that they are partially affected by unphysical spurious effects ? In other words is it reasonable to use them as w-references ?
- Line 335: Reanalyses are significantly affected by spin-up effects.
- Line 428: Not good due to spin-up.

- o Response for Lines 330-331, Line 335 and Line 428 comments. Thank you for raising this these points. Although the reanalyses present uncertainties, reanalyses such as GLORYS12 and ECCOv4 are widely used in the oceanographic community (e.g., Wunsch, 2011, Gray and Riser, 2014, Thomas et al., 2014, Liao et al., 2022). Additionally, the comparison between different datasets with different input sources and methodologies allows us to identify robust large-scale features that are consistently represented across products. This multi-dataset comparison provides a reliable baseline for validation, even in the presence of individual dataset uncertainties and spin-up artifacts. Furthermore, all these issues are an additional motivation for using OLIV3.

- Line 348-349: I don't understand this sentence. I would say that the geostrophic vertical velocity in the ocean interior results from the convergence/divergence of the Ekman drift.
  - o Thank you for the comment. As discussed in Section 2.1, the geostrophic vertical velocity at a given depth is determined by two contributions: the Ekman pumping at the ocean surface and the β-plane divergence of the geostrophic flow in the interior.

- Line 355: Figure 5. Sorry but I am lost with vertical gradient expressed in ms−1/kgm−3 . Where does σ27 fit in relation to σMLD ? I suggest expressing this gradient in day−1 = mday−1/m.
  - o Thank you for the suggestion. We have revised the calculation of the vertical gradient, now "diapycnal gradient" in Figure 6 (previous Figure 5) by using the vertical distance between isopycnal surfaces (in meters) rather than the difference in density (σ27 − σMLD), as you suggested. Only regions where the mixed-layer depth (MLD) is shallower than σ27 are considered, ensuring that the gradient reflects the vertical structure of the thermocline consistently. We have changed the figures and text according to the new metric.

- Line 375-376: Be careful w(z = 0)≠ wek.
  - o With the update of Section 2.1 and the addition of Appendix A, we are confident that this affirmation is correct.

- Line 387: Change Fig.5b to Fig.6b.
  - o Thank you for noticing it. Changed

- Line 391-393: Arbitrary conclusion at first glance (Fig 6). Make difference maps.
  - o We have included the difference map in the new Figure 8 to add robustness to our discussion.

- Line 446: OMEGA3D also integrates vertical stratification.
- Line 450: OMEGA3D is a physical investigating tool because it is based on the destruction of the thermal wind balance by current and turbulence.
  - o Thank you for these two comments. To clarify, we have updated the lines 487-488: "*In contrast, OMEGA3D employs the omega equation, which, although it also requires vertical integration, explicitly includes second-order vertical derivatives and horizontal derivatives of $w$.*"

- Line 527-530: If I am a biogeochemical scientist, or physicist who wants to estimate modal water production, what is the benefit of using OLIV3 rather than a reanalysis ? Sorry, I'm not convinced, but I would like to be.
  - Thank you very much for your question. OLIV3 provides a robust description of the vertical flow in the open-ocean large-scale subtropical downwellings and tropical upwellings, in particular its interannual variability. Therefore, if one desire to study more water formation variability, we recommend to use OLIV3 in priority to any other existing global product since w variability in mode water regions is likely completely dominated by its geostrophic component (if we trust the comparison showing very good correlation between wg and wtot in our reference OGCM, Figure 4b). Compared with reanalysis, OLIV3 is a tool based on observations without being affected by all the biases of reanalyses products (e.g. spin up effects), as the reviewer commented. While studies such as Bellacicco et al. (2025) use OMEGA3D to estimate the biological carbon pump, OLIV3 offers a complementary dataset with a more physically consistent vertical structure than OMEGA3D. Again, our aim is to provide the community with an additional variable computed from and independent input and a different methodology to what it is available for the community at the moment.
- Line 532: Before incorporating non-linear processes, integrate before the total meridional velocity and vorticity.
  - Changed "full" by "total" in line 578 to improve understanding.

In conclusion, I request substantial changes, particularly on the interest of using OLIV3, and clarifications on the incorporation of Ekman pumping in Equation 2, and the physical interpretation of this equation balance.

  - We thank the reviewer for the revision and for highlighting all these key points. In response, we have made substantial revisions to the manuscript to clarify the interest and added value of OLIV3, the relationship between Ekman pumping and the geostrophic vertical velocities at the ocean surface and interior, and the metrics used for the validation of the product. In addition to addressing the specific issues in their locations in the text, some changes have been applied to the abstract and the conclusions section to reflect them.
  - We also realised that the manuscript repeatedly referred to our previous study (Cortés-Morales and Lazar, 2024). To lighten the text, we now refer to this wors as CM24 throughout the manuscript.

---

## Author Comment (AC2)

**Review of "Global Thermocline Vertical Velocities: A Novel Observation Based Estimate" by Diego Cortés-Morales, Alban Lazar, Diana Ruiz Pino, and Juliette Mignot.**

In this manuscript, Cortés-Morales et al. develop and validate a new vertical velocity estimate for the ocean. The manuscript is generally well-written, and the topic is timely. I believe that providing a vertical velocity dataset is extremely important for the oceanographic community. I have a few comments that, while not dramatically changing the manuscript and its results, will require some revisions from the authors. I hope these will help clarify key aspects of the manuscript.

For these reasons, I recommend a major revision of the manuscript.

Please see my specific comments below.

We thank the reviewer for the revision and for highlighting all these key points. In response, we have made substantial revisions to the manuscript to clarify the relationship between Ekman pumping and the geostrophic vertical velocities at the ocean surface and interior, and the metrics used for the validation of the product. In addition to addressing the specific issues in their locations in the text, some changes have been applied to the abstract and the conclusions section to reflect them.

We also realised that the manuscript repeatedly referred to our previous study (Cortés-Morales and Lazar, 2024). To lighten the text, we now refer to this wors as CM24 throughout the manuscript.

**Major Comments:**

1) Ekman Boundary Condition

Ekman dynamics are not geostrophic in the traditional sense, as they involve other processes (i.e., wind stress, vertical momentum transfer, and viscosity) and do not include the pressure gradient. Thus, it is confusing to refer to a vertical velocity that incorporates such dynamics as "geostrophic". In my opinion, this is more complex. A purely geostrophic vertical velocity would be represented by (2), with wg(zref) determined solely by geostrophic processes (perhaps using $\partial_t\eta$, where $\eta$ is the sea surface height). I would understand that you do no want to alter your framework, which is reasonable; however, you need to clarify this point and use more accurate terminology.

In section 2.1, if one derives the full equation, it becomes evident that this condition must be imposed by "patching" the Ekman layer and the interior ocean. Therefore, the condition exists not at the ocean surface but at the base of the Ekman layer.

- Thank you very much for your comment. We have revised Section 2.1 and added an Appendix A to include a more detailed discussion of the relationship between geostrophic velocities and Ekman pumping, supported by literature review. This revision clarifies the implications of assuming the Ekman pumping at the base of the Ekman layer and why in our formulation the Ekman pumping is defined at the ocean surface.

The sentence on lines 106-107 contradicts what is discussed in section 3.5. If the Ekman condition governs wg, how can wg at the boundary differ from Ekman? Either there is an issue or something is unclear in your method.

In section 3.5, you begin by stating, "one might wonder how near-surface wg compares with Ekman pumping (wEk)", yet your methodological description suggests they are the same (i.e., surface boundary conditions). You are comparing at different depths. This entire paragraph, discussion, and section are confusing; I do not understand the goal or conclusion. More importantly, how is wEk defined at different depths? Should it not be defined at the base of the Ekman layer? Are you setting it to random depths for comparison? This does not make sense.

- Thank you for pointing out this potential inconsistency. We clarify four points:
  - First, at the surface boundary, we impose the condition $w_{\text{tot}}(z = 0) = 0$. It is exact for the stationary flow of the time-mean state we first show in the paper, but not for the time-varying flow of the annual mean flow with interannual variability. It is therefore an approximation to neglect $d(\eta)/dt$ ($\eta$=sea surface height relative to the mean state reference surface level), and the very good results we obtain (correlation in Fig. 4b) validates a posteriori the assumption. Then we assume that the ageostrophic flow is at first order only produced by surface wind stress and friction (Ekman three-dimensional flow) and note $w_{\text{tot}} = w_g + w_{ag}$. Hence, the above surface condition implies:, $w_g(z = 0) = -w_{ek}(z = 0)$.
  - Second, this boundary condition does not imply that the geostrophic vertical velocity is identical to the Ekman pumping throughout the upper ocean. Applied to the Ekman layer, the assumption in the ocean of incompressibility of the total flow implies:

    $div(\boldsymbol{u_g}) = -div(\boldsymbol{u_{Ek}})$
    where $\boldsymbol{u_g}$ and $\boldsymbol{u_{Ek}}$ are the geostrophic and Ekman components of the threedimensional velocity field.

    Integrated in z, it gives:
    $$\int_{D_{Ek}}^{0} -\frac{\beta v_g}{f} + \frac{\partial w}{\partial z} dz' = \int_{D_{Ek}}^{0} -\nabla \cdot \boldsymbol{u_{Ek}} \, dz'$$

$$\int_{D_{Ek}}^{0} -\frac{\beta v_g}{f} dz' + w_g(0) - w_g(D_{Ek}) = \int_{D_{Ek}}^{0} -\nabla \cdot \boldsymbol{u_{h,Ek}} \, dz' - w_{Ek}(0) + w_{Ek}(D_{Ek})$$

In this equation, $w_{Ek}(D_{Ek}) = 0$ (because at $D_{Ek}$ each component of the 3D Ekman flow, is dissipated/vanished), the surface condition eliminated $w_g(0)$ and $w_{Ek}(0)$, and Ekman theory gives

$\nabla \cdot \boldsymbol{U_{h,Ek}} = \nabla \times \left(\frac{\tau}{\rho_0 f}\right)$ for $\nabla \cdot \boldsymbol{U_{h,Ek}} = \int_{D_{Ek}}^{0} \nabla \cdot \boldsymbol{u_{h,Ek}} \, dz'$

Hence we obtain:

$$w_g(z = D_{Ek}) = \nabla \times \left(\frac{\tau}{\rho_0 f}\right) - \int_{D_{Ek}}^{0} \frac{\beta v_g}{f} \, dz'.$$

The magnitude of the vertical flow at the base of the Ekman layer is modified from its surface value by the vertically integrated divergence of the horizontal geostrophic flow over the Ekman layer. These corrections of the theory were recently proposed by Jacox et al., 2018. We understand "near-surface" may be confusing. We have changed "near-surface" "ocean interior" in lines 495-496:"...*one might wonder how ocean interior wg compares with surface Ekman pumping (wEk)."*

- o Third, regarding the definition and comparison of $w_{ek}$ and $w_g$ at different depths: in most of the literature (excepts for a few papers like Pedlosky, 1996 and Jacox et al., 2018), the Ekman pumping is often defined at the base of the Ekman layer , instead of the surface as demonstrated above. However, this the depth of that base varies regionally and among the different works. For clarity, due to the performance of OLIV3 in reproducing the temporal variability of the reference datasets, in Section 3.5 our intention is to compare the Ekman pumping signal ($w_{ek}(z = 0)$) with the geostrophic and total vertical velocity at selected upper-ocean levels in an OGCM simulation. This way, we propose an assessment of the error made when assuming that the wind stress derived Ekman pumping is an optimal proxy for the total vertical velocity variability within the upper ocean.
- o Finally, to test whether Ekman pumping alone explains upper-ocean variability, we compare correlations of the total vertical velocity with (a) surface Ekman pumping and (b) the geostrophic component evaluated at the same interior depth as the total field. The higher correlations of the geostrophic component at the corresponding level at which we evaluate the total velocity, indicate that variability of the total vertical velocity is not solely due to surface Ekman pumping, but also to interannual (and longer) variability in meridional geostrophic transport.
- o We have clarified these points in the text, section 2.1, appendix A and section 3.5. We hope that the review will better understand our arguments.

Lines 518-526: I find this paragraph confusing. Please rewrite it based on the comments above.

- • Thank you for highlighting this point. We have rewritten the paragraph in lines 566-567 to clarify that our intention is to examine the relationship between the surface

Ekman pumping and the vertical flow within the ocean interior: *"The wind-driven divergence at the surface and the vertical flow in the ocean interior are strongly correlated across large portions of the global tropical and subtropical gyres, as supported by the comparison between OGCM estimation of the geostrophic vertical velocities ($w_g$) in the ocean interior and Ekman pumping ($w_{Ek}$)."*

2) "Overuse" of the Correlation

Even if you demonstrate a good correlation, this does not imply good representation; it only indicates good synchrony. A correlation can still be good whereas one component is significantly stronger than the other. To fully illustrate good representation, you also need to check that the variances are consistent. Hence, if you want to demonstrate good representation, it would be more effective to use the coefficient of determination (R2: https://en.wikipedia.org/wiki/Coefficient_of_determination). I suggest to use instead of correlation for the particular purpose of your study.

Furthermore, correlation (normalized projection) and the coefficient of determination (R2; relative similarity = 1− relative error) are two distinct diagnostics. They can converge, but they do not have to. Please be specific and avoid conflating the two.

More importantly, how do you compute a local correlation with a "time mean" term (as indicated in the figure title 3b)? If w=fct(x,y,σ,t), the correlation is computed between two scalars... Projecting one scalar onto another does not make sense. The text references interannual correlation, thought, so please clarify!

- Thank you for this detailed and constructive comment. Our aim in this section is to quantify how much the time variability of the total vertical velocity can be attributed to its geostrophic component. We agree that correlation coefficient alone measures only the temporal synchrony between two variables and does not itself guarantee similarity in amplitude or variance. For this reason, our analysis does not rely solely on the correlation coefficient, we also examine the time-mean relative error between the two fields (Figs. 3a and 5), which provides complementary information about discrepancies in magnitude. Together, these diagnostics allow us to assess both the temporal coherence and the amplitude consistency of the geostrophic and total vertical velocity fields. In Figure 3c, we focus on how the geostrophic component describes the vertical structure of the total vertical velocity.

  Regarding the choice of metric, we believe that the correlation coefficient remains appropriate in this case. Unlike the coefficient of determination, the correlation field allows as to identify regions where the geostrophic and total vertical velocities are anticorrelated, as in the case of western boundary currents. This highlights areas where the ageostrophic processes dominate the variability.

  Concerning the reviewer's question about the "time-mean" in Figure 3 (now Figure 4): this was indeed a typographical error in the title of the subplots. The time-mean corresponds to panels (a) and (c), while panel (b) corresponds to interannual

(annual resolution) correlation between $w_g$ and $w_{tot}$. We have corrected the figure title.

We acknowledge the possible overuse of the correlation coefficient, and we clarify that results of these figures are only to test synchronicity, while a good representation also implies the time mean comparison in lines 266-267: "*To evaluate the ability of the geostrophic component to represent the spatiotemporal variability of the total vertical flow, we examine three diagnostics:*" and lines 329-332: "*Most notably, the high and widespread synchrony at annual frequencies between wtot and wg suggests that the geostrophic component strongly dominates the interannual variability, and therefore likely in the real ocean as well. While nonlinear processes influence the mean amplitude of the vertical flow, they play a smaller role in modulating this variability.*"

3) Description of the dataset

I believe there was a missed opportunity to present the dataset effectively. Adding a single figure that illustrates vertical velocity at a range of key levels (surface, within the thermocline, and below the thermocline) would be beneficial. A single figure with six eight panels showing vertical velocity in color and the corresponding isopycnal depth in contour would serve as the main visual aid to describe the dataset.

This figure, along with a descriptive paragraph, should be included in section 2.2.

- Thank you for your comment. We have included a new Figure 1 as the reviewer suggested showing OLIV3 at different isopycnal levels across the thermocline.

4) Two datasets (?)

Two datasets are provided and attached to the manuscript (isopycnal level and vertical level). However, only the isopycnal dataset is presented, described, and analyzed. I suggest removing the vertical-level dataset for consistency, as it seems surprising, to say the least, to include a supplementary dataset that is neither referenced nor discussed in the manuscript. Alternatively, a full description and analysis of the second dataset are required.

- Both datasets originate from the same computation, integrating LVB over the native ARMOR3D vertical levels. To address your concern, we have removed the vertical-level dataset and kept only the isopycnal interpolated one. We have revised lines 172-176: "*The resulting Observation-based LInear Vorticity Vertical Velocities (OLIV3) product consists of geostrophic vertical velocities derived from ARMOR3D meridional velocities and surface Ekman pumping from ERA5 wind stress, using the vertically integrated geostrophic LVB (Eq. \ref{eq2}) over the native ARMOR3D vertical levels and then interpolated over isopycnal levels. The product spans the 1993 - 2019 period, with a horizontal resolution of 0.25$^\circ$ and 71 isopycnal levels.*".

**Minor Comments:**

*) In the introduction (lines 54-58), you should also mention a strategy for reconstructing the full 3D circulation (u,v,w) based on thermal wind balance and Argo float deep displacement knowledge for the reference-level horizontal velocity. The vertical velocity derives similarly from a Sverdrup balance (equivalent to your approach): reference Colin de Verdière and Ollitrault (2016) and Colin de Verdière et al. (2023).

Ref:

Colin de Verdière, A., and M. Ollitrault, 2016. A Direct Determination of the World Ocean Barotropic Circulation. Journal of Physical Oceanography, 46, 255-273.

Colin de Verdière, A., T. Meunier and M. Ollitrault, 2019, Meridional overturning and heat transport from Argo floats displacements and the planetary geostrophic method: applications to the subpolar North Atlantic, Journal of Geophycical Research, 124, 6270-6285.

- Thank you for suggesting these references. We have updated the introduction (lines 45-59) to include a discussion of this additional strategy for reconstructing the full 3D circulation using thermal wind balance combined with Argo float displacements to define the reference-level horizontal velocity. This includes the explicit citation of Colin de Verdière and Ollitrault (2016) and Colin de Verdière et al. (2019) : *"Traditional approaches for estimating vertical velocities across different ocean regions were derived from tracer fluxes (e.g. Stommel and Arons, 1959; Robinson and Stommel, 1959; Wyrtki et al., 1961; Munk, 1966; Wunsch, 1984) or the application of the continuity equation to horizontal current measurement obtained from hydrographic station data (e.g. Stommel and Schott, 1977; Schott and Stommel, 1978; Wyrtki, 1981; Roemmich, 1983) and mooring measurements (e.g. Halpern and Freitag, 1987; Halpern et al., 1989; Weingartner and Weisberg, 1991; Helber and Weisberg, 2001). These early methods provided insight into the small vertical velocities' order of magnitude and upwelling/downwelling patterns of the vertical motions. Vertical velocities have also been inferred from the divergence of horizontal velocity in numerical models (e.g. Madec et al., 2019), although it remains impractical for global observation-based applications because of the sparse distribution of direct current measurements. Exceptions to such application to observations are Freeland (2013), which used in situ Argo float observations to estimate w at a single depth in a limited domain of several degrees, assuming zero vertical flow at the surface, and De Verdière and Ollitrault, 2016 and De Verdière et al., 2019, which computed vertical velocities from the horizontal divergence of geostrophic velocities inferred from Argo-derived three-dimensional thermohaline fields and float displacements at their parking depth. In the last decade, alternative approaches used isopycnal displacements (Giglio et al., 2013; Christensen et al., 2023), the use of mooring data combined with the momentum and density balances (Sevellec et al., 2015), and biogeochemical tracers (Garcia-Jove et al., 2022). The theoretical frameworks have also expanded to include methods based on the Bernoulli function to infer w (Tailleux et al., 2023)."*

*) In several places, "Naveira Garabato" has been referred to as "Garabato." Please revise this.

- Thank you for noticing. Changed in line 65 and reference list in lines 804 and 851.

*) Line 97: To the best of my knowledge, you do not need to assume the $\beta$-plane (i.e., tangent slope to sphere, $f$ varies linearly, or $\beta$ is constant). You only need to acknowledge that $\beta=\partial_y f$ (i.e., the meridional derivative of $f$ exists).

- Thank you for this clarification. What we intended to communicate is that our formulation requires that the meridional derivative of f exists locally. We have therefore revised the text accordingly in lines 100-101: "*This formulation is equivalent to the continuity equation of the geostrophic flow when considering a β-plane (Pedlosky, 1996), where β is the meridional derivative of the Coriolis parameter without requiring the classical assumption of a linear variation of f across the domain*"

*)After equation (2): Please add "assuming $f\neq0$." This clarification is important.

- Thank you for pointing out. Added in line 113.

*) Line 102: Please replace "geostrophic flow on a $\beta$-plane" with "horizontal geostrophic flow."

- Changed in line 114.

*) Line 122: This is the reference paper for the ANDRO dataset (deep displacements of Argo floats). I think that this is not be the correct reference, or I don't understand why you are citing it in this context.

- Thank you for pointing this out. You are correct: although the ARMOR3D product has been validated against the ANDRO dataset in previous studies, ANDRO is not used in the construction of ARMOR3D itself. Since the reference is not relevant in this context, we have removed it from the manuscript.

*) Sentences on lines 123-124: I do not find a reference for that. However, it is "admitted" in the community that this approach is not ideal. Have you tested your ability to execute this? How does the surface error propagate at depth? From my understanding, this is not as straightforward as it appears.

- Thank you for your comment. The statements on lines 159-167 are based on the works of Guinehut et al., 2012 and Mulet et al., 2012, which are cited at the beginning of the paragraph. We did not recompute the geostrophic velocities ourselves, we used the ARMOR3D product as provided for the community. Consequently, we did not perform additional testing of surface error propagation at depth, and our analysis assumes the quality and limitations reported in the product references.

Additionally, I am concerned that the boundary conditions for the vertical velocity are Ekman-based, while for horizontal flows, they are geostrophic. There exists a subtle but

fundamental difference between the two. Have you carefully checked this? For instance, is it dynamically consistent to have a horizontal flow at the surface that does not "see" to the wind, while a vertical flow derives from a horizontal flow that does?

- Thank you for raising this point. To compute vertical velocities from the LVB, a reference velocity is required. While level of no motion reference would have been an option, its uncertain location can introduce significant errors. Using the surface as reference therefore appear to us as the most appropriate choice. Concerning, the reviewer's point, at large scales, the geostrophic horizontal flow dominates the interior ocean dynamics, while large-scale surface winds modify the surface heigh patterns, which in turn are used to derive the geostrophic horizontal field. Within the theoretical framework presented in Section 2.1 and the appendix A, the only ageostrophic contribution considered is the Ekman component. This ageostrophic flow is confined to the Ekman layer and translates as Ekman pumping. As demonstrated theoretically, Ekman pumping can be defined at the ocean surface, while below the Ekman layer, the flow is predominantly geostrophic. Therefore, our choice of variables is dynamically consistent, in accordance with Pedlosky, 1996.

*) Line 230, equation (4): Why is there an absolute value? This seems illogical. Please make a simple difference. Furthermore, it would be mathematically more accurate to use \Delta w / \Delta \sigma on the left; the use of $\partial$ is mathematically ambiguous (i.e., unclear) and physically inconsistent (in terms of units).

- Thank you for this comment. We have revised Equation 4 (now Equation 13) to improve both notation and clarity. The absolute value of the vertical velocities is used in the definition of the vertical gradient to focus on the changes in amplitude within the thermocline, considering the positive and negative values could be misleading due to the existence of upward and downward vertical velocities across the domain. The revised equation and comments are updated in line 275: "*Negative values indicate a decrease in magnitude with depth, while positive values indicate an increase.*" We replaced the $\partial$ with \Delta_z.

*) Lines 284-293: Please specify the frequency of the vertical velocity (i.e., w_tot from the OGCM). If it is once every 5 days or lower, I am not surprised by these results. However, if it is hourly, I would be surprised that the vertical flow energy is not dominated by inertial motion, which would weaken the correlation with the geostrophic flow.

- Thank you for this comment. The OGCM provides vertical velocities at monthly frequency, as indicated in Table 1. However, for the correlation analysis presented in Section 3 we have used annual frequency following Cortés-Morales and Lazar, 2024. We have included "... *at annual frequencies* ..." in line 329 to avoid confusion on the temporal resolution of the analysis.

*) Figure 3: This does not represent a vertical gradient but rather a diapycnal gradient. Additionally, if you wish to show a relative term (which is debatable, since the denominator can be 0), you should use an absolute value in the denominator (otherwise, the sign becomes ambiguous, as the denominator can be both positive and negative).

- Thank you for your comment. You are correct: the quantity shown in Figure 3 (now Figure 4) represents the diapycnal gradient rather the vertical gradient. Following your suggestion, we have updated the units to m day$^{-1}$ and computed the gradient using the distance between isopycnal levels, yielding a diapycnal gradient in m day$^{-1}$ m$^{-1}$. Additionally, now use the absolute value of the diapycnal gradient in the denominator when calculating the relative error, to avoid ambiguity in sign. The relative error is intended to quantify the deviation of the geostrophic vertical velocity gradient from the total gradient.

*) In several places, "x" should be replaced by "$\times$" (e.g., line 303).

- Thank you for your comment. We have changed it.

*) Caption of Figure 7: Please replace "," with "," and replace "and" with ", and." *)

- Changed.

For Figures 9 and lines 463-465: Again, R2 would provide much more informative insights!

- Commented in Major Point (2).